# SECOND-ORDER FINE-TUNING WITHOUT PAIN FOR LLMS: A HESSIAN INFORMED ZEROTH-ORDER OPTIMIZER

**Yanjun Zhao**[1,*], **Sizhe Dang**[1,*], **Haishan Ye**[1,2,†], **Guang Dai**[2], **Yi Qian**[1,†], **Ivor W.Tsang**[3,4]

[1]Xi'an Jiaotong University, China, [2]SGIT AI Lab, State Grid Corporration of China, China
[3]CFAR and IHPC, Agency for Science, Technology and Research, Singapore
[4]College of Computing and Data Science, Nanyang Technological University, Singapore
{yanjun.zhao, darknight1118}@stu.xjtu.edu.cn
yehaishan@xjtu.edu.cn, yqian@mail.xjtu.edu.cn
guang.dai@gmail.com, ivor_tsang@cfar.a-star.edu.sg

## ABSTRACT

Fine-tuning large language models (LLMs) is necessary for specific downstream tasks, but the classic adaptive first-order optimizer entails prohibitive GPU memory because of backpropagation. Recent works such as MeZO have turned to zeroth-order optimizers for fine-tuning, which reduce substantial memory by using just two forward passes. However, heterogeneous curvatures across different parameter dimensions in LLMs often cause convergence instability or even failure. In this work, we propose HiZOO, a diagonal **H**essian **i**nformed **Z**eroth-**O**rder **O**ptimizer , which is the first to leverage the diagonal Hessian to enhance ZOO for fine-tuning LLMs. We provide the theoretical proof for HiZOO and visualize the optimization trajectories on the test functions. Extensive experiments on various models (RoBERTa, OPT, Phi-2, and LLama3, with 350M∼66B parameters) indicate that HiZOO significantly reduces the number of training steps and improves model accuracy. For example, on the SST2 task, HiZOO achieves an **8×** speed-up and better accuracy. Even when scaled to 66B-model, HiZOO outperforms MeZO with up to **5.1%** absolute improvement. We also propose HiZOO-L, which reduces the Hessian memory cost to **10%** of the MeZO, while maintaining almost same performance. Compared with ZO-Adam, HiZOO-L achieves a **4.3%** absolute improvement, just using **50%** of the GPU memory. Code is available at https://github.com/Yanjun-Zhao/HiZOO.

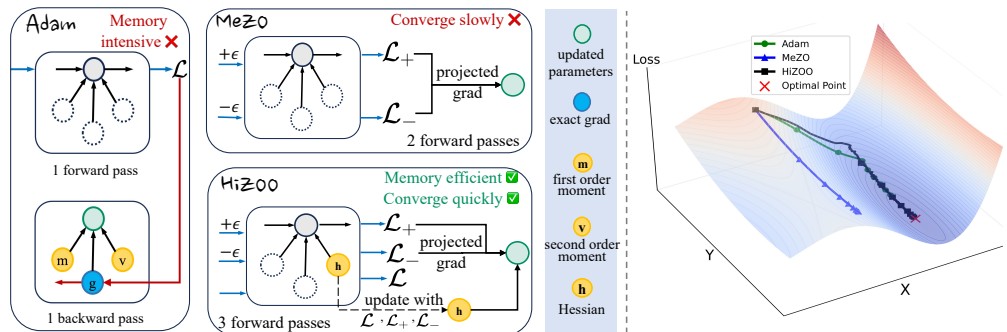

Figure 1: (Left) Comparison of HiZOO, MeZO and Adam. (Right) **Heterogeneous curvatures example**. HiZOO updates along the direction with greater curvature ($X$) and converges more quickly than MeZO. The corresponding loss curves are shown in Section 3.5.

---

*Equal contribution. This work was completed during the internship at SGIT AI Lab, State Grid Corporation of China. † Corresponding author.

## 1 INTRODUCTION

Fine-tuning pre-trained LLMs for specific tasks has gained significant attention recently. As the number of model parameters increases, full parameter fine-tuning (FT) becomes markedly memory-intensive. To alleviate GPU memory limitations, parameter-efficient fine-tuning (PEFT) methods (Hu et al., 2022; Li & Liang, 2021; Dettmers et al., 2023; Zhao et al., 2024b; Pan et al., 2024) have been developed, which only fine-tune a small number of (extra) model parameters. As a result, they significantly reduce the computational and storage cost, while achieving performance comparable to a fully fine-tuned model.

Adaptive first-order optimizers such as Adam (Kingma & Ba, 2015) and AdamW (Loshchilov & Hutter, 2019) are widely used to fine-tune LLMs. However, using these optimizers still leads to substantial memory consumption, primarily due to the inherent backpropagation process to calculate the gradient. To address these limitations, MeZO (Malladi et al., 2023) proposed to utilize a zeroth-order optimizer (ZOO) to estimate the gradient with just two forward passes per step, no need for backpropagation anymore. This achieves numerous memory reductions and makes it accessible to train and store LLMs on consumer hardware.

However, the parameters of LLMs often exhibit heterogeneous curvatures across different dimensions, as documented in recent studies (Sagun et al., 2017; Ghorbani et al., 2019; Zhang et al., 2020). This significant difference of second derivative makes the MeZO converge towards saddle point, slowing down the convergence speed, as shown in Figure 1 (right). Since the incorporation of Hessian to measure the curvature properties of the loss landscape, second-order methods (Liu & Li, 2023; Yao et al., 2021; Anil et al., 2021) can solve this suboptimal behavior. Unfortunately, in the context of zeroth-order optimization, one cannot directly compute the Hessian atop first-order derivatives.

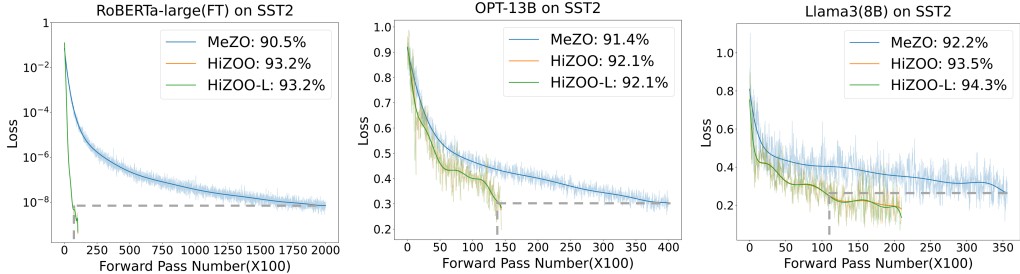

Figure 2: Performance of MeZO, HiZOO and HiZOO-L on SST2 task, when fine-tuning RoBERTa-large, OPT-13B, Llama3(8B) models. HiZOO can achieve $8\times$ speedup and 1.55% absolute accuracy improvement compared with MeZO.

In light of above, we propose HiZOO, as shown in Figure 1 (left), which estimates the diagonal Hessian by one more forward pass. HiZOO can act as a pre-conditioner, directly adjusting the update size of different parameters according to their curvatures. So that it can improve the model convergence when encountered with heterogeneous curvatures. As shown in Figure 2, HiZOO can significantly reduce number of training steps and improve model accuracy. Here we summarize our key contributions as follows:

1. In this work, we estimate the Hessian in zeroth-order optimizer to fine-tune LLMs for the first time. Our HiZOO reduces the total number of forward passes required for model convergence and achieves better accuracy. By utilizing diagonal Hessian, HiZOO reduces the corresponding memory cost from $\mathcal{O}(d^2)$ to $\mathcal{O}(d)$. Furthermore, we propose HiZOO-L, reducing the memory usage of Hessian to **10%** of the MeZO.

2. We provide theoretical analysis to prove that HiZOO provides an unbiased estimation of the Hessian. Also, we illustrate how HiZOO utilizes Hessian to improve the convergence process by visualizing the optimization trajectories on test functions.

3. We conduct extensive experiments across different models (RoBERTa-large, OPT, Llama3 and Phi-2) with scales from 350M to 66B, different methods (FT, LoRA, prefix), and different downstream tasks (classification, multiple-choice, and generation) to verify the effect of the HiZOO. For example, on SST2 task HiZOO achieves a better accuracy and

$8\times$ speedup over MeZO on average across different models. Even on OPT-66B, HiZOO outperforms better than MeZO with up to **5.1%** absolute improvement.

4. Further exploration in Section 4.3 showcases that HiZOO can achieve better performance in optimizing non-differentiable objectives such as F1 score. Specifically, HiZOO significantly outperforms MeZO 's results with **6.5%** absolute on average.

## 2 RELATED WORKS

Here we present a concise overview on optimizers used in fine-tuning LLMs(details in Appendix A).

**First-Order adaptive optimizer used in fine-tuning LLMs**   Optimization methods have consistently been a popular research domain. Adaptive first-order optimizer, such as Gradient Descent (GD), Momentum, Adagrad (Duchi et al., 2011), are fundamental in many areas like computer vision, natural languagle processing (NLP). Among them, Adam (Kingma & Ba, 2015) plays a dominant role due to its fast convergence and is often chosen for training and fine-tuning LLMs. AdamW (Loshchilov & Hutter, 2019) improves upon Adam by adding the weight decay to alleviate overfitting. But both of them requires lots of memory cost due to the backpropagation process. This issue has become increasingly critical as the number of LLM parameters skyrockets.

**Enhanced optimizers with Hessian**   On the other hand, researchers incorporated second-order information (Hessian) to provide richer guidance for gradient descent during the training. For example BROYDEN (BROYDEN, 1970) , Nesterov & Polyak (Nesterov & Polyak, 2006) and Conn et al. (Conn et al., 2000) utilized curvature information to pre-condition the gradient; Magoulas et al. (Magoulas et al., 1999) applied diagonal Hessian as the pre-conditioner; Martens (Martens, 2010) approximated the Hessian with conjugate gradient. Sophia (Liu & Li, 2023) used a light-weight estimate of the diagonal Hessian for pre-training LLMs. Despite their potential, above optimizers require the enormous GPU-memory cost. Additionally, these methods can only be used when first-order gradients are available.

**Zeroth-Order Optimizer**   Zeroth-order optimizers, with just forward passes to estimate the gradient, can greatly reduce the memory consumption. It appears in a wide range of applications where either the objective functions is implicit or its gradient is impossible or expensive to obtain. Methods like SPSA (Spall, 1992) have been shown to perform well in non-convex multi-agent optimization (Tang et al., 2021; Hajinezhad & Zavlanos, 2018) or generating black-box adversarial examples (Chen et al., 2017; Cai et al., 2021; Liu et al., 2019a; Ye et al., 2019). Recently, MeZO (Malladi et al., 2023) first adapted the classical ZO-SGD method to fine-tune LLM, achieving comparable performance with significant memory reduction. Then Zhang et al. (2024) proposed a wider array of ZO optimization techniques. However, these methods often struggle with heterogeneous curvatures.

## 3 METHODS

In the following, we briefly introduce the classical ZO gradient estimator SPSA (Spall, 1992), which is used in MeZO. Then we describe how HiZOO estimates diagonal Hessian and cooperates with ZOO. We also provide detailed proof for our method.

### 3.1 PRELIMINARIES

**Definition 3.1.** Simultaneous Perturbation Stochastic Approximation or SPSA

Given a model with parameters $\theta \in \mathbb{R}^d$ and loss function $\mathcal{L}$, SPSA estimates the gradient on a minibatch $\mathcal{B}$ , based on the concepts of sampling and differencing, as shown below:

$$g'_\mu(\theta_t) = \frac{\mathcal{L}(\theta_t + \mu u; \mathcal{B}) - \mathcal{L}(\theta_t - \mu u; \mathcal{B})}{2\mu} u \approx uu^\top \nabla \mathcal{L}(\theta_t; \mathcal{B}),$$

where $u \in \mathbb{R}^d$ and is sampled from $\mathcal{N}(0, I_d)$, $\mu$ is the *perturbation scale*. The $n$-SPSA gradient estimate averages $g_\mu(\theta)$ over $n$ randomly sampled $u$.

---

**Algorithm 1** HiZOO

---

**Require:** parameters $\theta \in \mathbb{R}^d$, loss $L : \mathbb{R}^d \to \mathbb{R}$, step budget $T$, perturbation scale $\mu$, learning rate schedule $\eta_t$, smooth scale $\alpha_t$, diagonal Hessian $\Sigma_0$

1: **for** $t = 1, ..., T$ **do**
2:      Sample batch $\mathcal{B} \subset \mathcal{D}$ and random seed $s$
3:      $\ell \leftarrow \mathcal{L}(\theta; \mathcal{B})$
4:      $\theta \leftarrow \text{PerturbParameters}(\theta, \mu, \Sigma_{t-1}^{1/2}, s)$
5:      $\ell_+ \leftarrow \mathcal{L}(\theta; \mathcal{B})$
6:      $\theta \leftarrow \text{PerturbParameters}(\theta, -2\mu, \Sigma_{t-1}^{1/2}, s)$
7:      $\ell_- \leftarrow \mathcal{L}(\theta; \mathcal{B})$
8:      $\theta \leftarrow \text{PerturbParameters}(\theta, \mu, \Sigma_{t-1}^{1/2}, s)$          ▷ Reset parameters before descent
9:      $\Sigma_t' = \frac{1}{2\mu^2}(\ell_+ + \ell_- - 2\ell)(\Sigma_{t-1}^{-1/2} u_i u_i^\top \Sigma_{t-1}^{-1/2})$          ▷ Update diagonal Hessian
10:      $\Sigma_t^{-1} = (1 - \alpha_t)\Sigma_{t-1}^{-1} + \alpha_t |diag(\Sigma_t')|$
11:      projected_grad $\leftarrow (\ell_+ - \ell_-) * \Sigma_t^{1/2}/2\mu$
12:      Reset random number generator with seed $s$          ▷ For sampling $u_i$
13:      **for** $\theta_i \in \theta$ **do**
14:          Sample $u_i \sim \mathcal{N}(0, I_d)$
15:          $\theta_i \leftarrow \theta_i - \eta_t * \text{projected\_grad} * u_i$
16:      **end for**
17: **end for**
18: **function** PERTURBPARAMETER($\theta, \mu, \Sigma_t^{1/2}, s$)
19:      Reset random number generator with seed $s$          ▷ For sampling $u_i$
20:      **for** $\theta_i \in \theta$ **do**
21:          Sample $u_i \sim \mathcal{N}(0, I_d)$
22:          $\theta_i \leftarrow \theta_i + \mu\Sigma_t^{1/2} u_i$          ▷ Modify parameters in place
23:      **end for**
24:      **return** $\theta$
25: **end function**

---

### 3.2   HESSIAN INFORMED ZEROTH-ORDER OPTIMIZATION

We will present how to estimate Hessian inverse matrix $\Sigma$ in detail in Section 3.3. Given $\Sigma$, then we can construct the following descent direction:

$$g_\mu(\theta_t) = \sum_{i=1}^{n} \frac{\mathcal{L}(\theta_t + \mu\Sigma_t^{1/2} u_i; \mathcal{B}) - \mathcal{L}(\theta_t - \mu\Sigma_t^{1/2} u_i; \mathcal{B})}{2\mu \cdot n} \cdot \Sigma_t^{1/2} u_i. \tag{1}$$

With the above descent direction, we can update $\theta_t$ as follows:

$$\theta_{t+1} = \theta_t - \eta_t g_\mu(\theta_t). \tag{2}$$

It's guaranteed that $g_\mu(\theta)$ can estimate the descent direction by the following equation:
$$\mathbb{E}\left[\mathcal{L}(\theta_{t+1}; \mathcal{B})\right] = \mathcal{L}(\theta_t; \mathcal{B}) - \eta_t \mathbb{E}\left[\langle \nabla\mathcal{L}(\theta_t; \mathcal{B}), g_\mu(\theta_t) \rangle\right] + \mathcal{O}(\eta_t^2)$$

$$= \mathcal{L}(\theta_t; \mathcal{B}) - \eta_t \frac{1}{b}\mathbb{E}\left[\sum_{i=1}^{b}\langle \nabla\mathcal{L}(\theta_t; \mathcal{B}), \Sigma_t^{1/2} u_i u_i^\top \Sigma_t^{1/2} \nabla\mathcal{L}(\theta_t; \mathcal{B}) \rangle\right] + \mathcal{O}(\eta_t^2) + \mathcal{O}(\mu)$$

$$= \mathcal{L}(\theta_t; \mathcal{B}) - \eta_t \|\Sigma_t^{1/2} \nabla\mathcal{L}(\theta_t; \mathcal{B})\|^2 + \mathcal{O}(\eta_t^2) + \mu,$$

where the first and second equality are both from the Taylor's expansion. Above equation shows that when $\eta_t$ is properly chosen, $g_\mu(\theta)$ can accurately estimate the direction of gradient descent, which is the key to the success of fine-tuning large language models.

### 3.3   DIAGONAL HESSIAN ESTIMATOR

Given a model with parameters $\theta \in \mathbb{R}^d$, storing the exact full spectral Hessian ($d \times d$) requires $\mathcal{O}(d^2)$ memory (Yao et al., 2018; Xu et al., 2019; Dembo et al., 1982), which is sufficient but never

necessary. In HiZOO, we just estimate and retain only the diagonal Hessian which requires $\mathcal{O}(d)$ memory. It serves as a pre-conditioner to scale the direction and magnitude of the model parameter updates according to their respective curvatures.

Drawing from the lemma presented in MiNES (Ye, 2023):

$$\frac{1}{2} \cdot \mathbb{E}_u(u^\top \Sigma^{1/2} H \Sigma^{1/2} u \cdot (\Sigma^{-1/2} u u^\top \Sigma^{-1/2} - \Sigma^{-1})) = H, \tag{3}$$

where $H$ is the Hessian $\nabla^2 \mathcal{L}(\theta; \mathcal{B})$ and $\Sigma$ is a positive definite matrix.

Thus, we can approximate the diagonal Hessian by the zeroth order oracles. Firstly, we will access to the $\mathcal{L}(\theta + \mu\Sigma^{1/2}u; \mathcal{B})$, $\mathcal{L}(\theta - \mu\Sigma^{1/2}u; \mathcal{B})$ and $\mathcal{L}(\theta; \mathcal{B})$. Through the Taylor's expansion, we yield the following results:

$$\mathcal{L}(\theta + \mu\Sigma^{1/2}u; \mathcal{B}) = \mathcal{L}(\theta; \mathcal{B}) + \mu\langle \mathcal{L}(\theta; \mathcal{B}), \Sigma^{1/2}u \rangle + \frac{\mu^2}{2} u^\top \Sigma^{1/2} \nabla^2 \mathcal{L}(\theta; \mathcal{B}) \Sigma^{1/2} u + \alpha(\theta, \mu\Sigma^{1/2}u).$$

Similarly, we also have:

$$\mathcal{L}(\theta - \mu\Sigma^{1/2}u; \mathcal{B}) = \mathcal{L}(\theta; \mathcal{B}) - \mu\langle \mathcal{L}(\theta; \mathcal{B}), \Sigma^{1/2}u \rangle + \frac{\mu^2}{2} u^\top \Sigma^{1/2} \nabla^2 \mathcal{L}(\theta; \mathcal{B}) \Sigma^{1/2} u + \alpha(\theta, -\mu\Sigma^{1/2}u).$$

Then we can calculate the difference $\Delta\mathcal{L}$ by:

$$\begin{aligned} \Delta\mathcal{L} &= \mathcal{L}(\theta + \mu\Sigma^{1/2}u; \mathcal{B}) + \mathcal{L}(\theta - \mu\Sigma^{1/2}u; \mathcal{B}) - 2\mathcal{L}(\theta; \mathcal{B}) \\ &= \mu^2 u^\top \Sigma^{1/2} \nabla^2 \mathcal{L}(\theta; \mathcal{B}) \Sigma^{1/2} u + \alpha(\theta, \mu\Sigma^{1/2}u) + \alpha(\theta, -\mu\Sigma^{1/2}u). \end{aligned}$$

Since $\alpha(\theta, \mu\Sigma^{1/2}u)$ and $\alpha(\theta, -\mu\Sigma^{1/2}u)$ are of order $\mathcal{O}(\mu^3)$, we can obtain that:

$$\frac{\Delta\mathcal{L}}{\mu^2} = u^\top \Sigma^{1/2} \nabla^2 \mathcal{L}(\theta; \mathcal{B}) \Sigma^{1/2} u + \mathcal{O}(\mu).$$

Upon substituting the above results into the left side of the Eq. equation 3, we arrive at:

$$\frac{1}{2}\mathbb{E}\left[\frac{\Delta\mathcal{L}}{\mu^2} \cdot \left(\Sigma^{-1/2} u u^\top \Sigma^{-1/2} - \Sigma^{-1}\right)\right] = \nabla^2 \mathcal{L}(\theta; \mathcal{B}) + \mathcal{O}(\mu).$$

Therefore, by generalizing above equation to the multi-sampling version, we can approximate the diagonal Hessian $\nabla^2 \mathcal{L}(\theta)$ at $\theta$ by:

$$\Sigma'_t(\theta) = \frac{1}{2n} \sum_{i=1}^n \left[\frac{\Delta\mathcal{L}}{\mu^2} \cdot \left(\Sigma_t^{-1/2} u_i u_i^\top \Sigma_t^{-1/2} - \Sigma^{-1}\right)\right], \tag{4}$$

where $n$ denotes the number of sampling instances for $u$, indicating the frequency of estimation per step. A larger $n$ diminishes the variance of the diagonal Hessian estimation and simultaneously increases computational overhead. Here we adopt $n = 1$ as the default setting and present the pseudo-code of HiZOO in Algorithm 1. Further experimental investigation into the impact of varying $n$ is available in the Section 4.6.

Above equation shows that we can approximate the diagonal entries of $\nabla^2 \mathcal{L}(\theta; \mathcal{B})$ by $\text{diag}(\Sigma'_t(\theta))$, requiring just one more forward pass per step compared with MeZO.

Due to the presence of noise in the calculation of the Hessian, we utilize exponential moving average (EMA) to denoise the diagonal Hessian estimation.

$$\Sigma_{t+1}^{-1} = (1 - \alpha_t)\Sigma_t^{-1} + \alpha_t \left|\text{diag}(\Sigma'_t)\right|. \tag{5}$$

In the above equation, we firstly initial the $\Sigma_0 = I_d$ and update it every step with $\mathcal{O}(d)$ memory cost all the time. We also use $\left|\text{diag}(\Sigma'_t)\right|$ to keep all entries of $\Sigma_t$ to be non-negative.

To further reduce Hessian memory consumption, we propose HiZOO-L to maintain it in a low-rank subspace, motivated by Adafactor (Shazeer & Stern, 2018). For $\hat{\Sigma}^{-1} \in \mathbb{R}^{p \times q}$, we will store two low-rank matrices $R \in \mathbb{R}^{p \times k}$ and $C \in \mathbb{R}^{k \times q}$ with $k = 1$. Specifically, we can get $\hat{\Sigma}^{-1}$ by:

$$\hat{\Sigma}_t^{-1} = (R_t * C_t)/(1_p^\top * R_t),$$

where $1_p = (1, \cdots, 1) \in \mathbb{R}^p$ denotes a column vector of $p$ ones. Then in each step, we will update the $R$ and $C$ separately:

$$R_t^{-1} = (1 - \alpha_t)R_{t-1}^{-1} + \alpha_t \left| diag(\hat{\Sigma}_t') \right| * 1_q,$$

$$C_t^{-1} = (1 - \alpha_t)C_{t-1}^{-1} + \alpha_t 1_p^\top * \left| diag(\hat{\Sigma}_t') \right|.$$

Detailed Algorithm can be seen in Appendix D.

### 3.4 CONVERGENCE ANALYSIS

In this section, we will analyse the convergence based on the assumption of non-convex optimization (details in Appendix B).

**Theorem 3.2.** *Let the descent direction $g_\mu(\theta_t)$ defined as:*

$$g_\mu(\theta_t) = \sum_{i=1}^{b} \frac{\boldsymbol{L}(\theta_t + \mu\Sigma_t^{1/2}u_i; \mathcal{B}_t) - \boldsymbol{L}(\theta_t - \mu\Sigma_t^{1/2}u_i; \mathcal{B}_t)}{2b\mu} \Sigma_t^{1/2}u_i. \tag{6}$$

*Based on Assumption B.1-B.3, if the update rule for $\theta$ is $\theta_{t+1} = \theta_t - \eta g_\mu(\theta_t)$ for a single step, then it's established that:*

$$\mathbb{E}\left[\boldsymbol{L}(\theta_{t+1})\,]\right] \leq \boldsymbol{L}(\theta_t) - \frac{\eta_t}{4}\|\nabla\boldsymbol{L}(\theta_t)\|_{\Sigma_t}^2 + 2\eta_t^2 L\left(\mathrm{tr}(\Sigma_t) + \beta_u\right)\sigma^2 + \mathcal{O}(\mu^2). \tag{7}$$

*Furthermore, given iteration number $T$, we choose the step size $\eta = \frac{1}{8\sqrt{T}L(\max_t \mathrm{tr}(\Sigma_t)+\beta_u)}$ and take $\theta_{out} = \theta_j$ with $j$ uniformly sampled from $\{1, \ldots, T\}$. Then, we have*

$$\mathbb{E}\left[\|\nabla\boldsymbol{L}(\theta_{out})\|^2\right] \leq \frac{32L\left(\max_t\{\mathrm{tr}(\Sigma_t)\} + \beta_u\right)\left(\boldsymbol{L}(\theta_1) - \boldsymbol{L}(\theta_*)\right)}{\sqrt{T}\beta_\ell} + \frac{\sigma^2}{T^{3/2}\beta_\ell} + \mathcal{O}\left(\mu^2\right), \tag{8}$$

*where $\boldsymbol{L}(\theta_*)$ minimizes the function $\boldsymbol{L}(\theta;)$. The above equation shows that as $T \to \infty$, HiZOO can converge to the stationary point.*

*Proof.* Detailed proof can be found in Appendix B. $\square$

### 3.5 VISUALIZATION OF HiZOO ON TEST FUNCTIONS

Despite above theoretical guarantee, **we still want to illustrate how HiZOO utilizes Hessian to improve the convergence process**. But it's impractical for large models to visualize their optimization trajectories. Therefore we choose three test functions (see details in Appendix C) with heterogeneous curvatures across different parameters and visualize the optimization trajectories on them.

As illustrated in Figure 3, HiZOO and Adam both achieve better convergence on three functions, and HiZOO even requires less steps for convergence than Adam. However, MeZO only achieves effective convergence in either the $x$ or $y$ dimension, but not both, indicating a limitation in capturing this curvature difference. Particularly in function (c) curvature of $x$ is extremely bigger than $y$. In this case, HiZOO can sense this difference in parametric curvature and update the function along $x$ on purpose, achieving quicker convergence. In contrast, MeZO is very hard to converge.

## 4 EXPERIMENTS

Large language models are generally classified into two types: (1) Encoder-Decoder, also known as masked language models, such as BERT (Devlin et al., 2019) and ALBERT (Lan et al., 2020); (2) Decoder-Only, also recognized as generative language models, such as GPT family (Radford et al., 2019; Brown et al., 2020), OPT (Zhang et al., 2022a), LLaMA (Touvron et al., 2023), Phi (Li et al., 2023; Gunasekar et al., 2023).

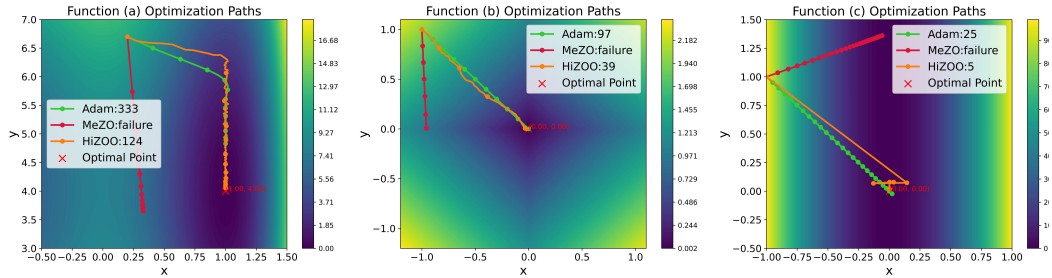

Figure 3: Optimization trajectories of Adam, MeZO and HiZOO on 3 test functions. We have labeled the number of iterations required for the loss to drop to 0.1.

Table 1: Experiments on RoBERTa-large (350M parameters, k=16). PEFT represents using LoRA and prefix and we report the best result of them. All reported numbers are averaged accuracy (standard deviation) across 5 runs.

| Task Type | SST-2 | SST-5 | SNLI | MNLI | RTE | TREC | Average |
|---|---|---|---|---|---|---|---|
| | — sentiment — | | — natural language inference — | | | — topic — | |
| Zero-shot | 79.0 | 35.5 | 50.2 | 48.8 | 51.4 | 32.0 | 49.5 |
| LP | 76.0 (±2.8) | 40.3 (±1.9) | 66.0 (±2.7) | 56.5 (±2.5) | 59.4 (±5.3) | 51.3 (±5.5) | 58.3 |
| FT | 91.9 (±1.8) | 47.5 (±1.9) | 77.5 (±2.6) | 70.0 (±2.3) | 66.4 (±7.2) | 85.0 (±2.5) | 74.9 |
| PEFT | 91.9 (±1.0) | 47.7 (±1.1) | 77.2 (±1.3) | 67.7 (±1.4) | 66.6 (±2.0) | 85.7 (±1.3) | 72.8 |
| MeZO | 90.5 (±1.2) | 45.5 (±2.0) | 68.5 (±3.9) | 58.7 (±2.5) | 64.0 (±3.3) | 76.9 (±2.7) | 67.4 |
| MeZO (PEFT) | 91.4 (±0.9) | 45.8 (±2.0) | 71.6 (±2.5) | 62.1 (±2.5) | 61.0 (±3.9) | 80.3 (±3.6) | 68.7 |
| HiZOO | **93.2** (±0.8) | 46.2 (±1.1) | **74.6** (±1.3) | **64.9** (±1.7) | **66.8** (±1.2) | 79.8 (±1.3) | **70.9** |
| HiZOO(PEFT) | 92.3 (±1.2) | **47.2** (±1.1) | 71.1 (±1.1) | 62.1 (±1.7) | 65.4 (±1.2) | **82.0** (±2.0) | 70.0 |

To rigorously assess the universality and robustness of our HiZOO, we have chosen models from each category for empirical testing. Additionally, we investigate FT and PEFT (LoRA (Hu et al., 2022) and prefix (Li & Liang, 2021)). Detailed experiment settings are presented in Appendix E.1.

## 4.1 MASKED LANGUAGE MODELS

Firstly, we conduct experiments on RoBERTa-large 350M (Liu et al., 2019b) on three NLP task paradigms: sentence classification, multiple choice and text generation. We follow the experimental setting (Malladi et al., 2023) in studying the few-shot and many-shot, sampling $k$ examples per class for $k = 16$ (results in Table 1) and $k = 512$ (results in Appendix E.1). We did not utilize HiZOO-L here due to model's smaller parameter count.

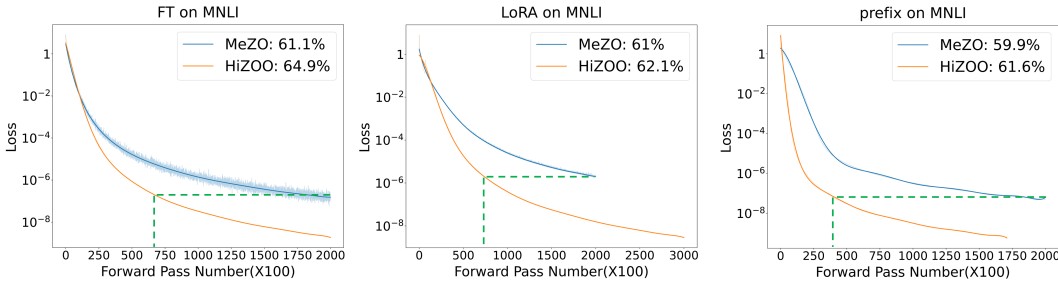

Figure 4: Training loss curves when using Adam, MeZO and HiZOO to fine-tune Roberta-large on MNLI. The evaluation accuracy curves can be found in Figure 11 in Appendix E.1.

**HiZOO greatly increases the convergence speed across full-parameter tuning, LoRA and prefix.** As shown in Figure 4, HiZOO achieves **4×** speedup over MeZO on average while getting the same training loss compared with MeZO. What's more, HiZOO finally achieves a **2.2%** absolute accuracy improvement on MNLI better than MeZO.

**HiZOO achieves better performance compared with MeZO.** Table 1 shows that HiZOO outperforms MeZO's results with **3.5%** absolute on average on all datasets across different tasks. Specifically, HiZOO outperforms MeZO more than **6%** in both the SNLI and MNLI dataset.

Table 2: Experiments on three different models(with 1000 examples). We highlight the best results between MeZO, HiZOO and HiZOO-L in bold to facilitate comparison.

| Model | Method | SST-2 | RTE | CB | WSC | WIC | COPA | MultiRC | Average |
|-------|--------|-------|-----|-----|------|-----|------|---------|---------|
| Phi-2 | MeZO | 86.6 | 67.1 | 75.0 | 59.6 | 54.4 | 86.0 | 78.2 | 72.4 |
| Phi-2 | HiZOO | 88.9 | 69.0 | 75.2 | 62.5 | 59.4 | 86.0 | 79.2 | **74.3** |
| Phi-2 | HiZOO-L | 88.9 | 68.9 | 75.2 | 62.4 | 59.2 | 86.0 | 79.2 | 74.2 |
| Llama3 | MeZO | 92.2 | 74.4 | 69.6 | 63.5 | 57.8 | 88.0 | 77.6 | 74.7 |
| Llama3 | HiZOO | 93.5 | 75.1 | 69.6 | 63.5 | 59.7 | 89.0 | 78.2 | **75.5** |
| Llama3 | HiZOO-L | 94.3 | 75.1 | 69.6 | 63.5 | 57.7 | 89.0 | 77.9 | 75.3 |
| OPT-13B | MeZO | 91.4 | 66.1 | 66.0 | 63.5 | 59.4 | 88.0 | 57.3 | 70.2 |
| OPT-13B | HiZOO | 92.1 | 69.3 | 69.6 | 63.5 | 59.4 | 89.0 | 61.3 | **72.1** |
| OPT-13B | HiZOO-L | 92.1 | 68.2 | 67.9 | 65.4 | 59.4 | 89.0 | 61.1 | 71.9 |

## 4.2 AUTO-REGRESSIVE LANGUAGE MODELS

Then we extend experiments with Phi-2(2.7B), Llama3(8B) and OPT family on the same NLP task paradigms. The results of the experiment in Table 2 show that HiZOO outperforms MeZO in most cases. Also, we can see that HiZOO-L has only a slight decrease in accuracy. We also provide relative loss curves to show the better convergence process of our HiZOO in Appendix E.2.

**HiZOO is capable of scaling to large models with up to 66B parameters, while preserving its exceptional performance.** As depicted in Table 3, on OPT-30B HiZOO outperforms MeZO with up to **2.9%** increase and **1.1%** increase on average. Even scaling to OPT-66B, HiZOO(prefix) still outperforms MeZO(prefix) with up to **5.1%** increase and **2.7%** increase on average.

## 4.3 TRAINING WITH NON-DIFFERENTIABLE OBJECTIVES

Our proposed HiZOO employs gradient estimation to update parameters, allowing for the use of non-differentiable objectives for training. Following the setting of MeZO (Malladi et al., 2023), we conduct extensive experiments using F1 as optimization objective. The results presented in Table 4 indicate that our method outperforms MeZO by **6.54%** absolute on F1 on average.

Table 3: Experiments on OPT-30B (we use FT and prefix-tuning, report the best of them) and OPT-66B (we use prefix-tuning).

| Task | SST-2 | RTE | WSC | WIC | Average |
|------|-------|-----|------|-----|---------|
| 30B MeZO | 90.6 | 66.4 | **63.5** | 59.1 | 69.9 |
| 30B HiZOO | **91.2** | **69.3** | **63.5** | 60.2 | **71.0** |
| 30B HiZOO-L | 91.1 | 68.9 | **63.5** | 59.8 | 70.8 |
| 66B MeZO | **93.6** | 66.4 | 57.7 | 58.6 | 69.0 |
| 66B HiZOO | **93.6** | **71.5** | **60.6** | **61.1** | **71.7** |
| 66B HiZOO-L | **93.6** | 71.0 | 60.3 | 60.9 | 71.4 |

Table 4: Experiments on non-differentiable optimization objectives (F1). For classification ($k = 512$), we use full-parameter tuning and for SQuAD (1,000 examples), we use prefix tuning.

| Model | RoBERTa-large (350M) | | | | OPT-13B |
|-------|-------|-------|------|------|---------|
| Task | SST-2 | SST-5 | SNLI | TREC | SQuAD |
| Zero-shot | 79.0 | 35.5 | 50.2 | 32.0 | 46.2 |
| MeZO | 92.7 | 48.9 | 82.7 | 68.6 | 78.5 |
| HiZOO | **94.9** | **52.9** | **83.1** | **90** | **83.21** |

## 4.4 MEMORY USAGE AND TIME EFFICIENCY ANALYSIS

**Memory Usage** As shown in Figure 5, HiZOO increases the memory usage compared to MeZO because of the storage of the diagonal Hessian(refer to Appendix F for detailed numbers). To further reduce memory consumption, we propose HiZOO-L, the low-rank implementation of HiZOO, motivated by Adafactor (Shazeer & Stern, 2018). Detailed Algorithm can be seen in Appendix D. As a result, HiZOO-L increases $< \mathbf{10\%}$ memory more than MeZO, while maintaining the original performance of HiZOO. Specifically, using the same GPUs, HiZOO-L allows for tuning a model that is 10 times larger than what is feasible with FT on average.

**Time Efficiency** We analyse the wall-clock time efficiencies and find that HiZOO and HiZOO-L spend $1.5\times$ time per step compared with MeZO, mainly from the extra forward pass, details in

Appendix G. However, HiZOO reduces total number of forward passes required for convergence. For example, HiZOO achieves a **8×** and **4×** speedup on SST2 and MNLI tasks.

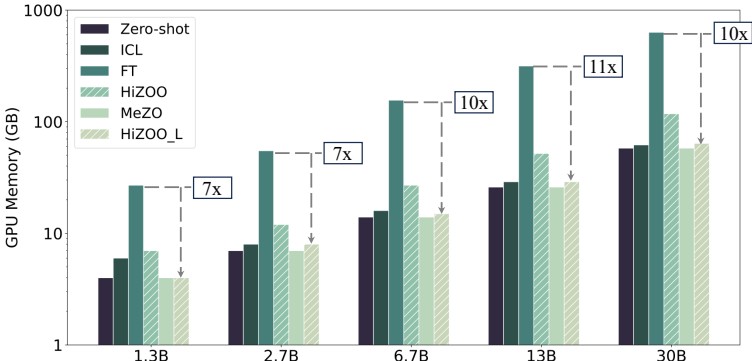

Figure 5: GPU memory consumption with different OPT models and tuning methods on MultiRC (400 tokens per example on average). More details can be found in Appendix F.

## 4.5 COMPARISON WITH OTHER ZO VARIANTS

We also compare our HiZOO with a broader array of ZO optimization techniques Zhang et al. (2024). As shown in Table 5, our HiZOO outperforms all other ZO methods. Compared with ZO-Adam who leverages second-order moment to guide gradient descent, our HiZOO-L achieves a notable **4.3%** absolute improvement, while using **50%** of the GPU memory.

Table 5: Performance comparison on SST2(Robert-Large and OPT-1.3B) and COPA(OPT-13B) using different ZO methods. Memory and runtime cost are multiples of ZO-SGD.

| Model/Task | Roberts-Large | | OPT-1.3B | | OPT-13B | | Average | Memory | Runtime |
|---|---|---|---|---|---|---|---|---|---|
| | FT | prefix | FT | prefix | FT | prefix | | | |
| ZO-SGD | 89.4 | 90.0 | **90.8** | 91.4 | **90.0** | 79.0 | 88.4 | 1.0x | 1.0x |
| ZO-SGD-MMT | 89.6 | 89.1 | 85.2 | 91.2 | 87.0 | 85.0 | 87.8 | 1.56x | 1.0x |
| ZO-SGD-Cons | 89.6 | 89.1 | 88.3 | 88.1 | 82.0 | 84.0 | 86.8 | 1.0x | 2.49x |
| ZO-SGD-Sign | 52.5 | 53.6 | 87.2 | 89.5 | 80.0 | 78.0 | 73.4 | 1.0x | 1.0x |
| ZO-Adam | 89.8 | 90.2 | 84.4 | **91.4** | 82.0 | 79.0 | 86.1 | 2.47x | 1.04x |
| HiZOO | **93.2** | **92.7** | 90.7 | **91.4** | 88.0 | **87.0** | **90.5** | 2.04x | 1.37x |
| HiZOO-L | 92.5 | **92.7** | 90.7 | **91.4** | 88.0 | **87.0** | 90.4 | 1.12x | 1.39x |

## 4.6 HYPERPARAMETER ANALYSIS

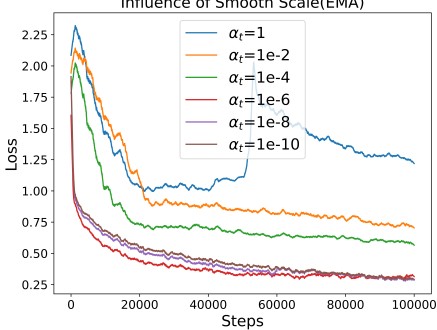

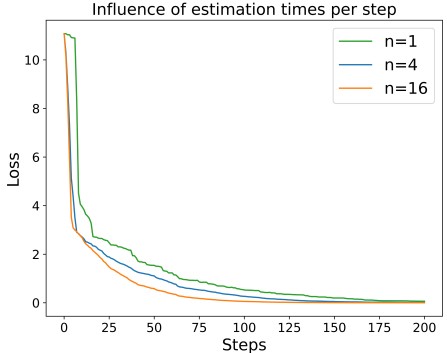

Figure 6: Influence of EMA $\alpha_t$ for hessian in Eq. equation 5. We use HiZOO (prefix) to fine-tune Roberta-large on SNLI. More results can be found in Appendix H.1.

Figure 7: Loss curves on Function (a) using the variant HiZOO-multi with different estimation times $n$ per step. Trajectory visualization can be found in Appendix H.1.

**Influence of Smooth Scale $\alpha_t$ in EMA** To assess the robustness of the optimizer, a grid search is conducted to evaluate the sensitivity of the hyper-parameter $\alpha_t$ on RoBERTa-large (350M). Figure 6 illustrates that as $\alpha_t$ is incrementally increased from zero, the training loss decreases faster. However, too large $\alpha_t$ values may impede convergence or even cause training to fail due to gradient explosion.

**Influence of Estimation Times $n$ Per Step** We also propose a variant of HiZOO in Appendix D.2: HiZOO-multi, which has $n > 1$ per step. As shown in Figure 7, different $n$ maybe doesn't affect the final accuracy. However, the larger $n$ will estimate the diagonal Hessian more accurate per step and accelerate model convergence, reducing the overall training steps. But it will also increase the computation per step. Balancing these factors is crucial for efficient training.

## 5 CONCLUSION

In this work, we introduce HiZOO, which is the first ZOO that incorporates diagonal Hessian for fine-tuning LLMs. By introducing one more forward pass, HiZOO can handle heterogeneous curvatures across different parameter dimensions. We provide theoretical analysis and visualize the optimization trajectories to explore how it works. Further experiments show that HiZOO converges in much fewer steps than MeZO and achieves better performance across various LLMs. We also explore a memory efficient implementation (HiZOO-L) to reduce the Hessian consumption.

## ACKNOWLEDGMENTS

This work was supported in part by National Key Research and Development Project of China under Grant 2022YFA1004002 and National Natural Science Foundation of China under Grant 72471185.

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

# A    RELATED WORKS

## A.1    FIRST-ORDER OPTIMIZER USED IN LLMS

Optimization methods have consistently been a popular research domain, encompassing techniques such as Gradient Descent (GD), Momentum, Adagrad (Duchi et al., 2011), ADADELTA (Zeiler, 2012), and Newton's method, which have been instrumental in advancing fields like computer vision. However, the emergence of large-scale models, characterized by their massive parameter counts and intricate architectures, has challenged the efficacy of conventional optimization methods for training tasks. Amidst this landscape, Adam (Kingma & Ba, 2015) has emerged as the preferred choice for its ability to rapidly converge, making it particularly suitable for the training and fine-tuning large models. Then AdamW (Loshchilov & Hutter, 2019) was proposed to add a weight decay coefficient to alleviate over-fitting. Notwithstanding these advancements, a limitation persists with these optimizers: they have an implicit batch size ceiling. Exceeding this threshold can provoke extreme gradient updates, thus impeding the convergence rate of the models. This bottleneck is particularly problematic in the context of large-model training, which typically necessitates substantial batch sizes. To circumvent this constraint, LAMB (You et al., 2020) was devised to apply principled layer-wise adaptation strategy to accelerate the training of large models employing large batches.

## A.2    HESSIAN BASED FIRST-ORDER OPTIMIZER

Compared with first-order optimizers, second-order optimizer considers second-order information in the process of gradient calculation. As a result, it has more abundant information to guide gradient descent and is considered to be more promising. Previous studies utilized curvature information to pre-condition the gradient (BROYDEN, 1970; Nesterov & Polyak, 2006; Conn et al., 2000). Subsequently, Magoulas et al. (Magoulas et al., 1999) applied diagonal Hessian as the pre-conditioner, which greatly promotes the landing of second-order optimizer in the field of deep learning. Martens (Martens, 2010) approximated the Hessian with conjugate gradient. Schaul et al. (Schaul et al., 2013) utilized diagonal Hessian to automatically adjust the learning rate of SGD during training. Another work (Pascanu & Bengio, 2014) extended natural gradient descent to incorporate second order information alongside the manifold information and used a truncated Newton approach for inverting the metric matrix instead of using a diagonal approximation of it. EVA (Zhang et al., 2023) proposed to use the Kronecker factorization of two small vectors to approximated the Hessian, which significantly reduces memory consumption. AdaHessian (Yao et al., 2021) incorporates an approximate Hessian diagonal, with spatial averaging and momentum to precondition the gradient vector.

Although great progress has been made in the research of second-order optimizer, it has not been widely used because of the extra computation and memory cost when gradient updating, and this situation is extremely serious in the training of large language models. Based on the above dilemma, recent works (Anil et al., 2021; George et al., 2018) proposed to offload Hessian computation to CPUs and utilized ResNets and very large batch size to approximate the Fisher information matrix. Sophia (Liu & Li, 2023) was the first to apply second-order optimizer and achieve a speed-up on large language models in total compute successfully.

## A.3    ZEROTH-ORDER OPTIMIZER

Zeroth-order optimization, is also known as derivative-free or black-box optimization. There have been many one-point gradient estimators in past works (FairScale authors, 2021; Spall, 1997; Vakhitov et al., 2009; Spall, 1992; Jamieson et al., 2012; Agarwal et al., 2009; Raginsky & Rakhlin, 2011; Wang et al., 2020). However, cursory experiments with one such promising estimator (Zhang et al., 2022b) reveal that SPSA outperforms other methods.

In previous works, it appears in a wide range of applications where either the objective function is implicit or its gradient is impossible or too expensive to compute. For example, methods (Tang et al., 2021; Hajinezhad & Zavlanos, 2018) consider derivative-free distributed algorithms for non-convex multi-agent optimization. ZO-BCD(Cai et al., 2021), ZOO(Chen et al., 2017), ZO-signSGD (Liu et al., 2019a) and ZO-HessAware (Ye et al., 2019) utilize zeroth-order stochastic optimization to generate black-box adversarial example in deep learning.

Beyond that, MeZO (Malladi et al., 2023) firstly adapted the classical ZO-SGD method to fine-tune LLMs, while achieving comparable performance with extremely great memory reduction and GPU-hour reduction. Recently there have been many subsequent excellent works (Jiang et al., 2023; Zhao et al., 2024a; Liu et al., 2024; Guo et al., 2024; Tang, 2024; Chen et al., 2024; Wang et al.; Chen et al., 2025a; Tan et al., 2025; Chen et al., 2025b; Sun et al., 2025). All of these optimizers provide researchers with a new and promising technique for fine-tuning large models.

## B  DETAILED CONVERGENCE ANALYSIS

Firstly, our convergence analysis requires the following assumptions:

**Assumption B.1.** The objective function $\boldsymbol{L}(\theta)$ is $L$-smooth, which means that for any $\theta_1, \theta_2 \in \mathbb{R}^d$, it holds that:

$$\boldsymbol{L}(\theta_2) \leq \boldsymbol{L}(\theta_1) + \langle \nabla \boldsymbol{L}(\theta_1), \theta_2 - \theta_1 \rangle + \frac{L}{2} \|\theta_2 - \theta_1\|^2. \tag{9}$$

**Assumption B.2.** The stochastic gradient $\nabla \boldsymbol{L}(\theta)$ has $\sigma^2$ variance, which means:

$$\mathbb{E}\left[\|\nabla \boldsymbol{L}(\theta) - \nabla \boldsymbol{L}(\theta)\|^2\right] \leq \sigma^2. \tag{10}$$

**Assumption B.3.** Each entry of $\Sigma_t$ lies in the range $[\beta_\ell, \beta_u]$ with $0 < \beta_\ell \leq \beta_u$.

Then we will give the detailed proof for convergence.

*Proof.* By the update rule of $\theta_t$ and Assumption B.1, we have

$$
\begin{aligned}
&\mathbb{E}\left[\boldsymbol{L}(\theta_{t+1}) \mid \theta_t\right] \\
\leq &\boldsymbol{L}(\theta_t) - \eta_t \mathbb{E}\left[\langle \nabla \boldsymbol{L}(\theta_t), g_\mu(\theta_t) \rangle\right] + \frac{L\eta_t^2}{2} \mathbb{E}\left[\|g_\mu(\theta_t)\|^2\right] \\
\leq &\boldsymbol{L}(\theta_t) - \eta_t \|\nabla \boldsymbol{L}(\theta_t)\|_{\Sigma_t}^2 + \eta_t \mathcal{O}\left(\mu \|\nabla \boldsymbol{L}(\theta_t)\|\right) \\
&+ 2\eta_t^2 L \left(\operatorname{tr}(\Sigma_t) + \beta_u\right) \|\nabla \boldsymbol{L}(\theta_t)\|_{\Sigma_t}^2 \\
&+ 2\eta_t^2 L \left(\operatorname{tr}(\Sigma_t) + \beta_u\right) \sigma^2 + \mathcal{O}(\mu^2) \\
\leq &\boldsymbol{L}(\theta_t) - \frac{\eta_t}{2} \|\nabla \boldsymbol{L}(\theta_t)\|_{\Sigma_t}^2 + 2\eta_t^2 L \left(\operatorname{tr}(\Sigma_t) + \beta_u\right) \|\nabla \boldsymbol{L}(\theta_t)\|_{\Sigma_t}^2 \\
&+ 2\eta_t^2 L \left(\operatorname{tr}(\Sigma_t) + \beta_u\right) \sigma^2 + \mathcal{O}(\mu^2) \\
= &\boldsymbol{L}(\theta_t) - \frac{\eta_t}{2} \left(1 - 4\eta_t L(\operatorname{tr}(\Sigma_t) + \beta_u)\right) \|\nabla \boldsymbol{L}(\theta_t)\|_{\Sigma_t}^2 \\
&+ 2\eta_t^2 L \left(\operatorname{tr}(\Sigma_t) + \beta_u\right) \sigma^2 + \mathcal{O}(\mu^2) \\
\leq &\boldsymbol{L}(\theta_t) - \frac{\eta_t}{4} \|\nabla \boldsymbol{L}(\theta_t)\|_{\Sigma_t}^2 + 2\eta_t^2 L \left(\operatorname{tr}(\Sigma_t) + \beta_u\right) \sigma^2 + \mathcal{O}(\mu^2),
\end{aligned}
$$

where the second inequality is because of Lemma B.4 and the last inequality is because of the value of $\eta_t$.

Rearrange above equation and summing up it, we can obtain that

$$
\begin{aligned}
\mathbb{E}\left[\sum_{t=1}^T \frac{\eta_t}{4} \|\nabla \boldsymbol{L}(\theta_t)\|_{\Sigma_t}^2\right] \leq &\sum_{t=1}^T \left(\boldsymbol{L}(\theta_t) - \boldsymbol{L}(\theta_{t+1})\right) \\
&+ 2\eta_t^2 L \left(\operatorname{tr}(\Sigma_t) + \beta_u\right) \sigma^2 + \mathcal{O}(T\mu^2) \\
= &\boldsymbol{L}(\theta_1) - \boldsymbol{L}(\theta_{T+1}) + 2\eta_t^2 L \left(\operatorname{tr}(\Sigma_t) + \beta_u\right) \sigma^2 + \mathcal{O}(T\mu^2) \\
\leq &\boldsymbol{L}(\theta_1) - \boldsymbol{L}(\theta_*) + 2\eta_t^2 L \left(\operatorname{tr}(\Sigma_t) + \beta_u\right) \sigma^2 + \mathcal{O}(T\mu^2).
\end{aligned}
$$

By taking $\theta_{\text{out}} = \theta_j$ with $j$ uniformly sampled from $\{1, \ldots, T\}$ and taking expectation, we can obtain that

$$
\mathbb{E}\left[\|\nabla \boldsymbol{L}(\theta_{\text{out}})\|^2\right] = \frac{1}{T}\sum_{t=1}^{T}\|\nabla \boldsymbol{L}(\theta_t)\|^2 \leq \frac{1}{T\beta_\ell}\sum_{t=1}^{T}\|\nabla \boldsymbol{L}(\theta_t)\|_{\Sigma_t}^2
$$

$$
\leq \frac{4(\boldsymbol{L}(\theta_1) - \boldsymbol{L}(\theta_*))}{T\beta_\ell\eta} + \frac{8\eta L\left(\text{tr}(\Sigma_t) + \beta_u\right)}{T\beta_\ell}\sigma^2 + \mathcal{O}(\mu^2)
$$

$$
= \frac{32L\left(\text{tr}(\Sigma_t) + \beta_u\right)(\boldsymbol{L}(\theta_1) - \boldsymbol{L}(\theta_*))}{\sqrt{T}\beta_\ell} + \frac{\sigma^2}{T^{3/2}\beta_\ell} + \mathcal{O}\left(\mu^2\right),
$$

where the first inequality is because of the assumption that the diagonal entries of $\Sigma_t$ is no less than $\beta_\ell$, $\qquad\square$

Eq. equation 7 shows that once we choose the step size $\eta$ properly, $\boldsymbol{L}(\theta_{t+1})$ will be less than $\boldsymbol{L}(\theta_t)$ in expectation up to some noises of order $\mu^2$. Specifically, if set $\eta = \frac{1}{8\sqrt{T}L(\max_t \text{tr}(\Sigma_t) + \beta_u)}$, Eq. equation 8 implies that we can find an solution $\theta_{\text{out}}$ such that $\mathbb{E}\left[\|\nabla \boldsymbol{L}(\theta_{\text{out}})\|^2\right] \leq \epsilon^2$ in $\mathcal{O}(\epsilon^{-4})$ iterations. This rate matches the one of (Ghadimi & Lan, 2013).

**Lemma B.4.** *We assume that Assumption B.2 and Assumption B.3 hold. Then, $g_\mu(\theta_t)$ defined in Eq. equation 6 has the following properties:*

$$
\mathbb{E}[g_\mu(\theta_t)] = \Sigma_t \nabla \boldsymbol{L}(\theta_t) + \mathcal{O}(\mu)
$$

$$
\mathbb{E}\left[\|g_\mu(\theta_t)\|^2\right] \leq 4\left(\text{tr}(\Sigma_t) + \beta_u\right)\|\nabla \boldsymbol{L}(\theta_t)\|_{\Sigma_t}^2
$$

$$
+ 4\beta_u\left(\text{tr}(\Sigma_t) + \beta_u\right)\sigma^2 + \mathcal{O}(\mu^2).
$$

*Proof.* By the definition of $g_\mu(\theta_t)$, we have

$$
g_\mu(\theta_t)
$$

$$
= \sum_{i=1}^{b} \frac{\boldsymbol{L}(\theta_t + \mu\Sigma_t^{1/2}u_i) - \boldsymbol{L}(\theta_t - \mu\Sigma_t^{1/2}u_i)}{2b\mu}\Sigma_t^{1/2}u_i
$$

$$
= \sum_{i=1}^{b} \frac{2\mu\nabla^\top \boldsymbol{L}(\theta_t)\Sigma_t^{1/2}u_i + \mathcal{O}(\mu^2)}{2b\mu}\Sigma_t^{1/2}u_i
$$

$$
= \frac{1}{b}\sum_{i=1}^{b}\Sigma_t^{1/2}u_i u_i^\top \Sigma_t^{1/2}\nabla \boldsymbol{L}(\theta_t) + \mathcal{O}(\mu).
$$

Thus, we can obtain that

$$
\mathbb{E}[g_\mu(\theta_t)] = \Sigma_t \nabla \boldsymbol{L}(\theta_t) + \mathcal{O}(\mu). \tag{11}
$$

Moreover,

$$
\mathbb{E}\left[\|g_\mu(\theta_t)\|^2\right]
$$

$$
= \mathbb{E}\left[\|\frac{1}{b}\sum_{i=1}^{b}\Sigma_t^{1/2}u_i u_i^\top \Sigma_t^{1/2}\nabla \boldsymbol{L}(\theta_t) + \mathcal{O}(\mu)\|^2\right]
$$

$$
\leq 2\mathbb{E}\left[\|\frac{1}{b}\sum_{i=1}^{b}\Sigma_t^{1/2}u_i u_i^\top \Sigma_t^{1/2}\nabla \boldsymbol{L}(\theta_t)\|^2\right] + \mathcal{O}(\mu^2)
$$

$$
\leq \frac{2}{b}\sum_{i=1}^{b}\mathbb{E}\left[\|\Sigma_t^{1/2}u_i u_i^\top \Sigma_t^{1/2}\nabla \boldsymbol{L}(\theta_t)\|^2\right] + \mathcal{O}(\mu^2)
$$

$$
= 2\text{tr}(\Sigma_t) \cdot \nabla^\top \boldsymbol{L}(\theta_t)\Sigma_t \nabla \boldsymbol{L}(\theta_t)
$$

$$
+ 2\nabla^\top \boldsymbol{L}(\theta_t)\Sigma_t^2 \nabla \boldsymbol{L}(\theta_t) + \mathcal{O}(\mu^2)
$$

$$
\leq 2\left(\text{tr}(\Sigma_t) + \beta_u\right)\nabla^\top \boldsymbol{L}(\theta_t)\Sigma_t \nabla \boldsymbol{L}(\theta_t) + \mathcal{O}(\mu^2),
$$

where the last equality is because of Lemma B.5.

Finally, we have

$$
\begin{aligned}
&\mathbb{E}\left[\nabla^\top \boldsymbol{L}(\theta_t)\Sigma_t\nabla\boldsymbol{L}(\theta_t)\right] = \mathbb{E}\left[\|\nabla\boldsymbol{L}(\theta_t)\|_{\Sigma_t}^2\right]\\
\leq& 2\mathbb{E}\left[\|\nabla\boldsymbol{L}(\theta_t)\|_{\Sigma_t}^2\right] + 2\mathbb{E}\left[\|\nabla\boldsymbol{L}(\theta_t)-\nabla\boldsymbol{L}(\theta_t)\|_{\Sigma_t}^2\right]\\
\leq& 2\|\nabla\boldsymbol{L}(\theta_t)\|_{\Sigma_t}^2 + 2\beta_u\mathbb{E}\left[\|\nabla\boldsymbol{L}(\theta_t)-\nabla\boldsymbol{L}(\theta_t)\|^2\right]\\
\leq& 2\|\nabla\boldsymbol{L}(\theta_t)\|_{\Sigma_t}^2 + 2\beta_u\sigma^2,
\end{aligned}
$$

where the second inequality is because of Assumption B.3 and the last inequality is because of Assumption B.2.

Therefore,

$$
\mathbb{E}\left[\|g_\mu(\theta_t)\|^2\right] \leq 4\left(\operatorname{tr}(\Sigma_t)+\beta_u\right)\|\nabla\boldsymbol{L}(\theta_t)\|_{\Sigma_t}^2 + 4\beta_u\left(\operatorname{tr}(\Sigma_t)+\beta_u\right)\sigma^2.
$$

$\square$

**Lemma B.5.** *(Magnus et al., 1978) Let $A$ and $B$ be two symmetric matrices, and $u$ obeys the Gaussian distribution, that is, $u \sim N(0, I_d)$. Define $z = u^\top A u \cdot u^\top B u$. The expectation of $z$ is:*

$$
\mathbb{E}_u[z] = (\operatorname{tr}A)(\operatorname{tr}B) + 2\operatorname{tr}(AB). \tag{12}
$$

## C  TEST FUNCTIONS OF THE OPTIMIZATION TRAJECTORIES

For better illustrating how HiZOO utilizes hessian to improve the convergence process, we choose below three test functions with heterogeneous curvatures across different parameters. In Figure 8, we provide the 2D convergence paths of three functions and the variation of their losses with respect to steps.

- Function (a): $f(x, y) = 8(x-1)^2(1.3x^2 + 2x + 1) + 0.5(y-4)^2$
- Function (b): $f(x, y) = |x| + |y|$
- Function (c): $f(x, y) = 10000x^2 + y^2$

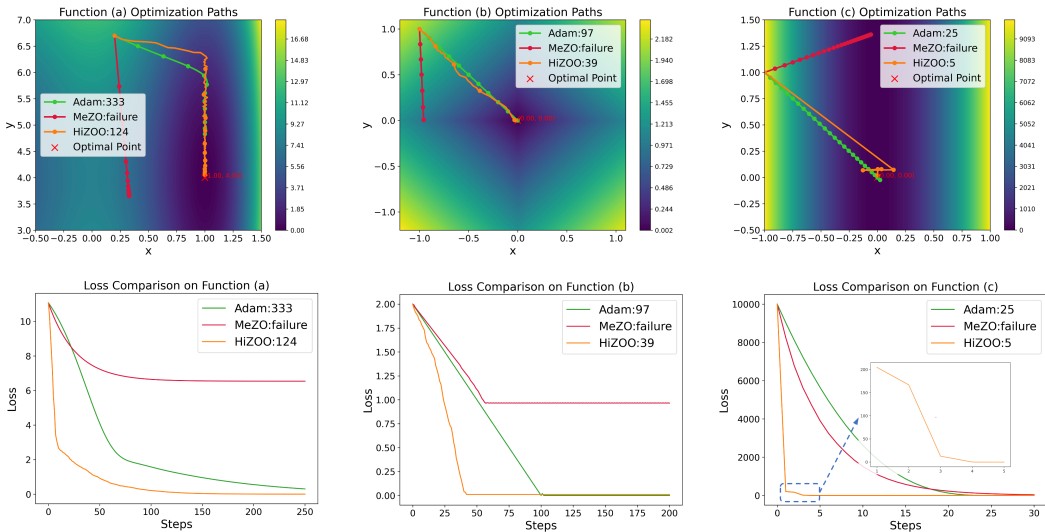

Figure 8: 2D trajectories of Adam, MeZO and HiZOO on 3 test functions. The upper figures are the 2D trajectories of gradient descent, and the bottom parts are the corresponding loss curves.

---

Function (a) is from (Liu & Li, 2023).

# D HiZOO Variants

## D.1 HiZOO-L

Due to the storage of Hessian, HiZOO introduces extra memory cost, which is equal to the size of the model parameters. To address this limitations, we propose HiZOO-L, the low-rank implementation for the storage of Hessian, motivated by Adafactor (Shazeer & Stern, 2018). Details can be see in Algorithm 2. We also visualize the loss curves of HiZOO and HiZOO-L in Figure 9 and find that on most datasets two algorithms perform closely. This also indicates that the estimation of Hessian in HiZOO may be sparse, so we encourage researchers to try other memory efficient algorithms to compress the Hessian.

---

**Algorithm 2** HiZOO-L

---

**Require:** parameters $\theta \in \mathbb{R}^d$, loss $L : \mathbb{R}^d \to \mathbb{R}$, step budget $T$, perturbation scale $\mu$, learning rate schedule $\eta_t$, smooth scale $\alpha_t$, diagonal Hessian $R_0, C_0$

1: **for** $t = 1, ..., T$ **do**
2:      Sample batch $\mathcal{B} \subset \mathcal{D}$ and random seed $s$
3:      $\ell \leftarrow \mathcal{L}(\theta; \mathcal{B})$
4:      $\theta \leftarrow \text{PerturbParameters}(\theta, \mu, R_{t-1}, C_{t-1}, s)$
5:      $\ell_+ \leftarrow \mathcal{L}(\theta; \mathcal{B})$
6:      $\theta \leftarrow \text{PerturbParameters}(\theta, -2\mu, R_{t-1}, C_{t-1}, s)$
7:      $\ell_- \leftarrow \mathcal{L}(\theta; \mathcal{B})$
8:      $\theta \leftarrow \text{PerturbParameters}(\theta, \mu, R_{t-1}, C_{t-1}, s)$          ▷ Reset parameters before descent
9:      $\hat{\Sigma}_{t-1}^{-1} = (R_{t-1} * C_{t-1})/(1_n^\top * R_{t-1})$
10:     $\hat{\Sigma}_t' = \frac{1}{2\mu^2}(\ell_+ + \ell_- - 2\ell)(\hat{\Sigma}_{t-1}^{-1/2} u_i u_i^\top \hat{\Sigma}_{t-1}^{-1/2})$
11:     $R_t^{-1} = (1 - \alpha_t)R_{t-1}^{-1} + \alpha_t \left|diag(\hat{\Sigma}_t')\right| * 1_m$
12:     $C_t^{-1} = (1 - \alpha_t)C_{t-1}^{-1} + \alpha_t 1_n^\top * \left|diag(\hat{\Sigma}_t')\right|$
13:     projected_grad $\leftarrow (\ell_+ - \ell_-) * \hat{\Sigma}_t^{1/2}/2\mu$
14:     Reset random number generator with seed $s$          ▷ For sampling $u_i$
15:     **for** $\theta_i \in \theta$ **do**
16:        Sample $u_i \sim \mathcal{N}(0, I_d)$
17:        $\theta_i \leftarrow \theta_i - \eta_t * \text{projected\_grad} * u_i$
18:     **end for**
19: **end for**

20: **function** PerturbParameter$(\theta, \mu, R_t, C_t, s)$
21:     Reset random number generator with seed $s$          ▷ For sampling $u_i$
22:     **for** $\theta_i \in \theta$ **do**
23:        Sample $u_i \sim \mathcal{N}(0, I_d)$
24:        $\hat{\Sigma}_t^{-1} = (R_t * C_t)/(1_n^\top * R_t)$
25:        $\theta_i \leftarrow \theta_i + \mu\hat{\Sigma}_t^{1/2} u_i$          ▷ Modify parameters in place
26:     **end for**
27:     **return** $\theta$
28: **end function**

---

## D.2 HiZOO-multi

There is a rich history of transferring ideas from first order optimization to enhance ZO algorithms. Below, we highlight the variant of HiZOO: HiZOO-multi which can perform $n$ estimation times per step efficiently as shown in Algorithm 3. We conducted experiments to explore the influence of estimation times $n$ per step as shown in Figure 16. We can conclude that when $n$ is larger, the estimation of diagonal Hessian is more accurate. It can decrease the variance of the estimated diagonal Hessian matrix during each step and thus reduce the overall training steps, but will cause much more computation per step meanwhile. So choosing an appropriate value of $n$ is very important during the training.

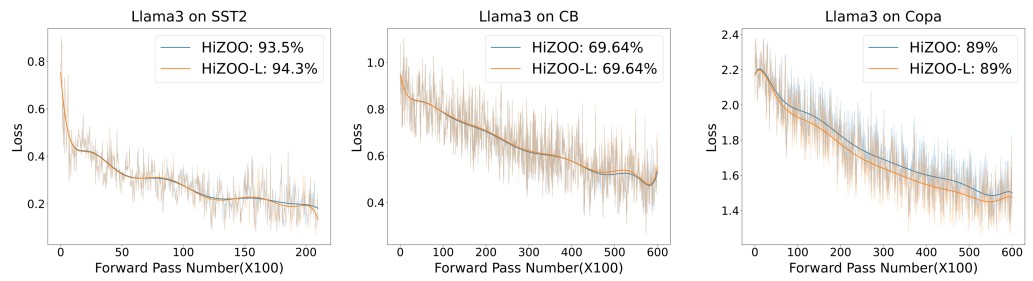

Figure 9: Loss curves on Llama3 between HiZOO and HiZOO-L.

---

**Algorithm 3** HiZOO-multi

---

**Require:** parameters $\theta \in \mathbb{R}^d$, loss $L : \mathbb{R}^d \to \mathbb{R}$, step budget $T$, perturbation scale $\mu$, batch size $B$, learning rate schedule $\eta_t$, smooth scale $\alpha_t$, estimate times $n$, Hessian matrix $\Sigma_0$

1: **for** $t = 1, ..., T$ **do**
2:     seeds, projected_grads $\leftarrow$ []
3:     **for** $j = 1, ..., n$ **do**
4:         Sample batch $\mathcal{B} \subset \mathcal{D}$ and random seed $s$
5:         $\ell \leftarrow \mathcal{L}(\theta; \mathcal{B})$
6:         $\theta \leftarrow$ PerturbParameters($\theta$, $\mu$, $\Sigma_{t-1}^{1/2}$, $s$)
7:         $\ell_+ \leftarrow \mathcal{L}(\theta; \mathcal{B})$
8:         $\theta \leftarrow$ PerturbParameters($\theta$, $-2\mu$, $\Sigma_{t-1}^{1/2}$, $s$)
9:         $\ell_- \leftarrow \mathcal{L}(\theta; \mathcal{B})$
10:        $\theta \leftarrow$ PerturbParameters($\theta$, $\mu$, $\Sigma_{t-1}^{1/2}$, $s$)        $\triangleright$ Reset parameters before descent
11:        $\Sigma_t' = \frac{1}{2\mu^2}(\ell_+ + \ell_+ - 2\ell)(\Sigma_{t-1}^{-1/2} u_i u_i^\top \Sigma_{t-1}^{-1/2})$
12:        $\Sigma_t^{-1} = (1 - \alpha_t)\Sigma_{t-1}^{-1} + \alpha_t |diag(\Sigma_t')|$        $\triangleright$ Update Hessian matrix
13:        projected_grad $\leftarrow (\ell_+ - \ell_-) * \Sigma_t^{1/2}/2\mu$
14:        projected_grads[j] $\leftarrow$ projected_grad
15:        seeds[j] $\leftarrow s$
16:     **end for**
17: **end for**
18: **for** $j = 1, ..., n$ **do**
19:     Reset random number generator with seeds[j]
20:     **for** $\theta_i \in \theta$ **do**
21:         $u_i \sim \mathcal{N}(0, I_d)$
22:         $\theta_i \leftarrow \theta_i - (\eta_t/n)*$projected_grads[j]$*u_i$        $\triangleright$ Avg grad for $u_1, ..., u_n$
23:     **end for**
24: **end for**

25: **function** PERTURBPARAMETER($\theta$, $\mu$, $\Sigma_t^{1/2}$, $s$)
26:     Reset random number generator with seed $s$
27:     **for** $\theta_i \in \theta$ **do**
28:         $u_i \sim \mathcal{N}(0, I_d)$
29:         $\theta_i \leftarrow \theta_i + \mu\Sigma_t^{1/2}u_i$        $\triangleright$ Modify parameters in place
30:     **end for**
31:     **return** $\theta$
32: **end function**

---

# E  EXPERIMENTS ON LLMS

## E.1  DETAILED EXPERIMENTS ON ROBERTA-LARGE

We use the hyperparameters in Table 6 for HiZOO experiments on RoBERTa-large. Regarding learning rate scheduling and early stopping, we use constant learning rate for all HiZOO experiments.

Table 6: The hyperparameter grids used for RoBERTa-large experiments. HiZOO uses a constant learning rate schedule. All HiZOO experiments use 100K steps.

| Experiment | Hyperparameters | Values |
|---|---|---|
| HiZOO | Batch size | 64 |
| | Learning rate | $\{1e-7, 1e-6, 1e-5\}$ |
| | $\mu$ | $1e-3$ |
| | Weight Decay | 0 |
| HiZOO(prefix) | Batch size | 64 |
| | Learning rate | $\{1e-2, 5e-3, 1e-3\}$ |
| | $\mu$ | $1e-1$ |
| | Weight Decay | 0 |
| | # prefix tokens | 5 |
| HiZOO(LoRA) | Batch size | 64 |
| | Learning rate | $\{1e-5, 5e-5, 1e-4\}$ |
| | $\mu$ | $1e-3$ |
| | Weight Decay | 0.1 |
| | $(r, \alpha)$ | $(8, 16)$ |

Table 7: Experiments on RoBERTa-large (350M parameters, k=512). For MeZO we report the results we reproduced.

| Task Type | SST-2 | SST-5 | SNLI | MNLI | RTE | TREC | Average |
|---|---|---|---|---|---|---|---|
| | —— sentiment —— | | —— natural language inference —— | | | — topic — | |
| Zero-shot | 79.0 | 35.5 | 50.2 | 48.8 | 51.4 | 32.0 | 49.5 |
| LP | 91.3 ($\pm$0.5) | 51.7 ($\pm$0.5) | 80.9 ($\pm$1.0) | 71.5 ($\pm$1.1) | 73.1 ($\pm$1.5) | 89.4 ($\pm$0.5) | 76.3 |
| FT | 91.9 ($\pm$1.8) | 47.5 ($\pm$1.9) | 77.5 ($\pm$2.6) | 70.0 ($\pm$2.3) | 66.4 ($\pm$7.2) | 85.0 ($\pm$2.5) | 73.1 |
| FT (LoRA) | 91.4 ($\pm$1.7) | 46.7 ($\pm$1.1) | 74.9 ($\pm$4.3) | 67.7 ($\pm$1.4) | 66.1 ($\pm$3.5) | 82.7 ($\pm$4.1) | 71.6 |
| FT (prefix) | 91.9 ($\pm$1.0) | 47.7 ($\pm$1.1) | 77.2 ($\pm$1.3) | 66.5 ($\pm$2.5) | 66.6 ($\pm$2.0) | 85.7 ($\pm$1.3) | 72.6 |
| MeZO | 93.3 ($\pm$0.7) | 53.2 ($\pm$1.4) | 83.0 ($\pm$1.0) | 78.3 ($\pm$0.5) | 78.6 ($\pm$2.0) | 94.3 ($\pm$1.3) | 80.1 |
| MeZO (LoRA) | 90.5 ($\pm$0.6) | 45.4($\pm$1.1) | 64.6($\pm$1.2) | 62.1($\pm$0.9) | 61.1($\pm$1.8) | 80.8($\pm$1.5) | 67.4 |
| MeZO (prefix) | 93.3 ($\pm$0.1) | 53.6 ($\pm$0.5) | 82.9 ($\pm$1.1) | 75.6 ($\pm$1.2) | 77.2 ($\pm$0.8) | 88.2 ($\pm$0.7) | 78.4 |
| HiZOO | 95.5 ($\pm$0.4) | 53.2 ($\pm$0.9) | 82.6 ($\pm$0.7) | 77.7 ($\pm$0.6) | **80.0** ($\pm$ 1.5) | **94.6** ($\pm$1.1) | 80.6 |
| HiZOO (LoRA) | 91.7 ($\pm$0.3) | 45.3 ($\pm$0.7) | 76.5 ($\pm$0.3) | 63.1 ($\pm$0.6) | 70.4 ($\pm$1.4) | 85.6 ($\pm$1.5) | 72.1 |
| HiZOO (prefix) | **96.1** ($\pm$0.2) | **54.2** ($\pm$0.4) | **85.7** ($\pm$0.7) | **79.7** ($\pm$1.0) | 77.3 ($\pm$0.2) | 93.9 ($\pm$0.6) | **81.2** |

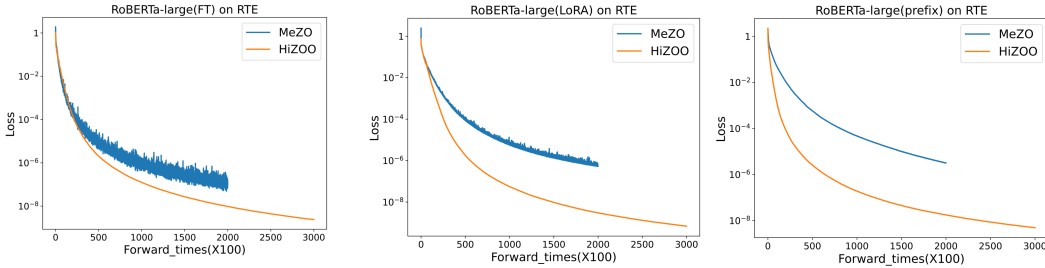

Figure 10: Loss curves on RoBERTa-large between MeZO and HiZOO.

In Table 7 we show the full experiment results. Additionally, we plot more loss curves to compare with MeZO. As shown in Figure 10, we can see that HiZOO can greatly accelerate the training process over MeZO, which verifies the robustness of HiZOO.

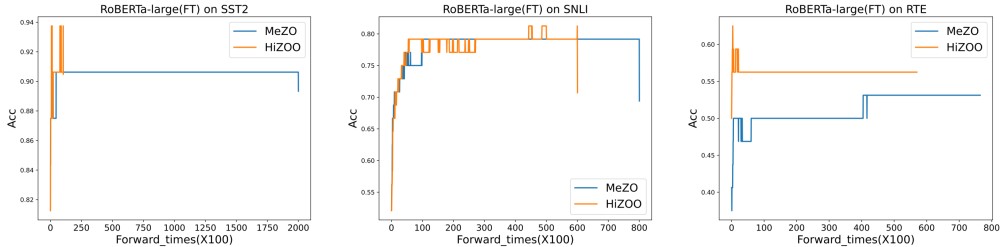

Figure 11: Accuracy curves on RoBERTa-large between MeZO and HiZOO.

## E.2 DETAILED RESULTS ON VARIOUS LLMs

We use the hyperparameters in Table 8 for HiZOO experiments on OPT. Full results for OPT-30B and OPT-66B are in Table 9. **We also provide the relative loss curves of fine-tuning OPT family in Figure 12.** We provide several loss curves of fine-tuning Phi-2(2.7B) and Llama3(8B) in Figure 13 and Figure 14.

Table 8: The hyperparameter grids used for OPT experiments. All weight decay is set to 0. HiZOO uses 20K steps and constant learning rates.

| Experiment | Hyperparameters | Values |
|---|---|---|
| HiZOO | Batch size | 16 |
| | Learning rate | $\{1e-6, 5e-7, 1e-7\}$ |
| | $\mu$ | $1e-3$ |
| HiZOO(prefix) | Batch size | 16 |
| | Learning rate | $\{5e-2, 1e-2, 5e-3\}$ |
| | $\mu$ | $1e-1$ |
| | # prefix tokens | 5 |
| HiZOO(LoRA) | Batch size | 16 |
| | Learning rate | $\{1e-4, 5e-5, 1e-5\}$ |
| | $\mu$ | $1e-2$ |
| | $(r, \alpha)$ | $(8, 16)$ |
| FT with Adam | Batch size | 8 |
| | Learning Rates | $\{1e-5, 5e-5, 8e-5\}$ |

Table 9: Experiments on OPT-30B and OPT-66B(with 1000 examples). The best results are highlighted in bold for better comparison. We highlight the best results between HiZOO and MeZO in bold to facilitate comparison.

| Task | SST-2 | RTE | WSC | WIC |
|---|---|---|---|---|
| 30B zero-shot | 56.7 | 52.0 | 38.5 | 50.2 |
| 30B ICL | 81.9 | 66.8 | 56.7 | 51.3 |
| 30B MeZO | 90.6 | 66.4 | **63.5** | 48.9 |
| 30B MeZO(prefix) | 87.5 | 66.1 | 55.8 | 59.1 |
| 30B HiZOO | 90.3 | **69.3** | **63.5** | 53.4 |
| 30B HiZOO(prefix) | **91.2** | 68.6 | 57.7 | **60.2** |
| 66B zero-shot | 57.5 | 67.2 | 43.3 | 50.6 |
| 66B ICL | 89.3 | 65.3 | 52.9 | 54.9 |
| 66B MeZO(prefix) | **93.6** | 66.4 | 57.7 | 58.6 |
| 66B HiZOO(prefix) | **93.6** | **71.5** | **60.6** | **61.1** |

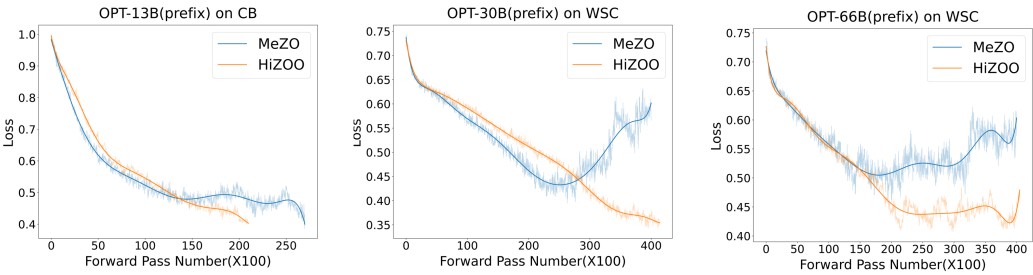

Figure 12: Loss curves on OPT between MeZO and HiZOO.

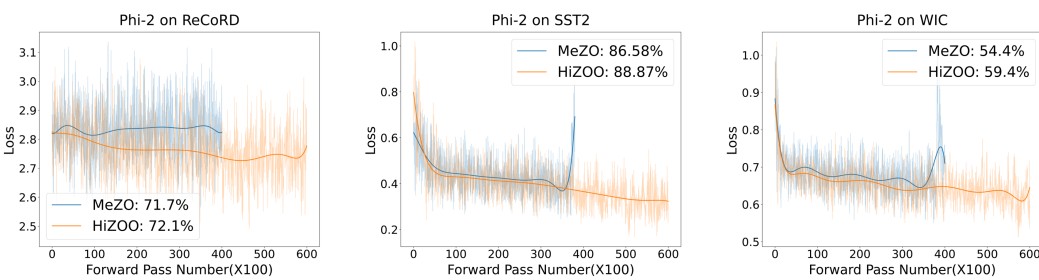

Figure 13: Loss curves on Phi-2 between MeZO and HiZOO.

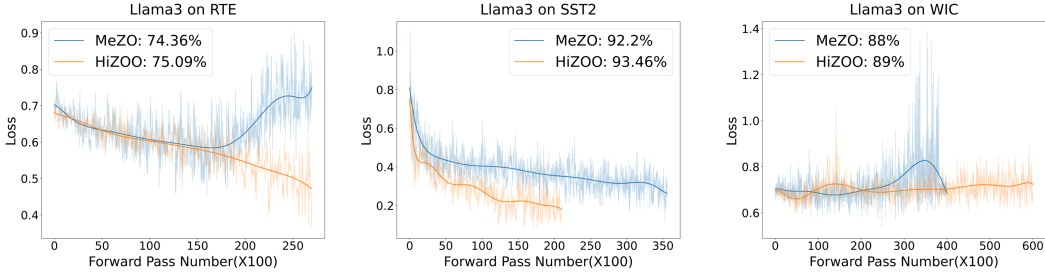

Figure 14: Loss curves on Llama3 between MeZO and HiZOO.

## F DETAILS ABOUT MEMORY USAGE

Here we show the detailed numbers of memory profiling results Table 10. We did not turn on any advance memory-saving options, e.g., gradient checkpointing. We set the per-device batch size as 1 to test the minimum hardware requirement to run the model with specific optimization algorithms. We use Nvidia's $nvidia-smi$ command to monitor the GPU memory usage.

Table 10: Memory usage on the MultiRC (average tokens=400) dataset. Results of ICL and full-parameter tuning are from MeZO(Malladi et al., 2023).

| Method | zero-shot/MeZO(FT) | HiZOO(FT) | HiZOO-L(FT) | ICL | Adam(FT) |
|--------|--------------------|-----------|-------------|-----|----------|
| 1.3B | 1xA100 (4GB) | 1xA100 (7GB) | 1xA100 (4GB) | 1xA100 (6GB) | 1xA100 (27GB) |
| 2.7B | 1xA100 (7GB) | 1xA100 (13GB) | 1xA100 (8GB) | 1xA100 (8GB) | 1xA100 (55GB) |
| 6.7B | 1xA100 (14GB) | 1xA100 (29GB) | 1xA100 (15GB) | 1xA100 (16GB) | 2xA100 (156GB) |
| 13B | 1xA100 (26GB) | 1xA100 (53GB) | 1xA100 (29GB) | 1xA100 (29GB) | 4xA100 (316GB) |
| 30B | 1xA100 (58GB) | 2xA100 (118GB) | 1xA100 (64GB) | 1xA100 (62GB) | 8xA100 (633GB) |
| 66B | 2xA100 (128GB) | 3xA100 (246GB) | 2xA100 (140GB) | 2xA100 (134GB) | 16xA100 |

## G DETAILS ABOUT WALLCLOCK TIME EFFICIENCY

In this section, we measure the wallclock time efficiency of HiZOO compared to MeZO and full-parameter fine-tuning (FT) with respect to different model sizes. Due to the lack of NV-Link

connectivity in our A100 GPUs, we selected models that can be fully fine-tuned on a single A100 GPU for comparison. As shown in Table 11, HiZOO exhibits a longer per-step duration compared to MeZO, within a 50% margin. This result indicates that the primary overhead in hierarchical optimization methods lies in the forward propagation process. Given that HiZOO involves an additional forward pass compared to MeZO, the time per step increases by approximately 1.4 to 1.5 times.

In conclusion, the speedup factors derived from the forward pass step used in our comparisons between HiZOO and MeZO reflect the actual wallclock time efficiency improvements accurately.

Table 11: Wallclock time per step between MeZO, HiZOO and HiZOO-L. The increase in wallclock time per step for HiZOO compared to MeZO is less than 1.5 times across different model sizes. To avoid introducing additional overheads such as inter-GPU communication, results are measured on the same dataset (SST-2) and GPUs (80GB A100), with each result averaged over 100 steps. "BS" refers to batch size. For the relatively smaller RoBERTa-large model, we used a BS=64, while for models larger than 1B parameters, we used a BS=16.

| Model | RoBERTa-large(350M) | Phi-2(2.7B) | Llama3(8B) | OPT(13B) |
|---|---|---|---|---|
| MeZO | 0.2092s(BS=64) | 0.3011s(BS=16) | 0.7471s(BS=16) | 1.1108s(BS=16) |
| HiZOO | 0.3023s(BS=64) | 0.4486s(BS=16) | 1.1090s(BS=16) | 1.5225s(BS=16) |
| HiZOO-L | 0.3193s(BS=64) | 0.4851s(BS=16) | 1.1996s(BS=16) | 1.6422s(BS=16) |

## H  DETAILS ABOUT ABLATION EXPERIMENTS

### H.1  INFLUENCE OF SMOOTH SCALE $\alpha_t$ AND NUMBER OF ESTIMATION $n$ PER STEP

We conducted experiments on SST-2, SST-5, MNLI datasets when fine-tuning RoBERTa-large to research the influence of smooth scale $\alpha_t$. Figure 15 shows that the value of $\alpha_t$ mainly affects the convergence speed of the model. Additionally, the best value of $\alpha_t$ will vary between different datasets. Figure 16 shows that the influence of the number of estimation $n$ per steps.

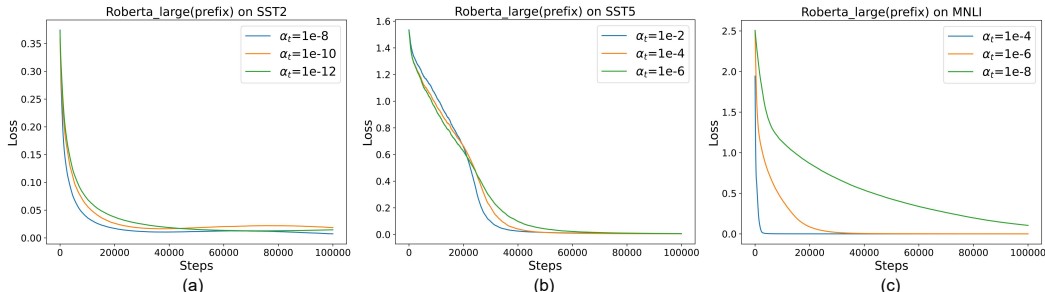

Figure 15: More experiments on influence of the value of Smooth scale $\alpha_t$ on RoBERTa.

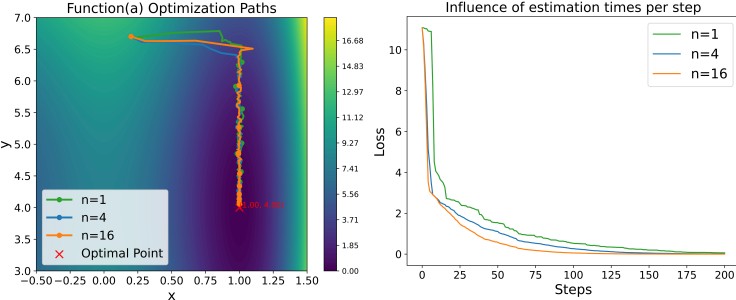

Figure 16: Influence of number of estimation per step. (left) 2D trajectories of gradient descent; (right) Corresponding loss curves.

## H.2   Experiments about Omitting $[-\Sigma^{-1}]$ term in Eq. equation 4

We conducted experiments on SST-2 datasets using three methods to fine-tune RoBERTa-large to compare the difference between with $[-\Sigma^{-1}]$ term and without this term. Figure 17 shows that this term can make negligible influence.

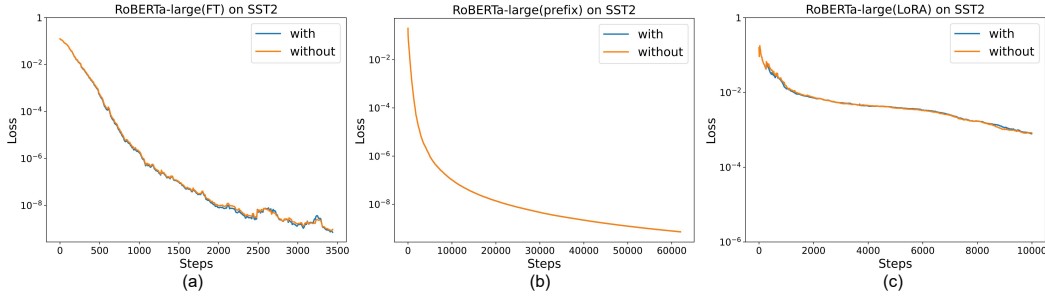

Figure 17: Experiment about the error generate by omitting the $[-\Sigma^{-1}]$ term in Eq equation 4. 'with' means holding the term and 'without' means omitting the term.

