# OpenReview forum: "Second-Order Fine-Tuning without Pain for LLMs: A Hessian Informed Zeroth-Order Optimizer"
_ICLR.cc/2025/Conference — ICLR 2025 Poster_

### Official Review · Reviewer_Wjn7 · 2024-11-04

**Soundness:** 3
**Presentation:** 2
**Contribution:** 2
**Rating:** 6
**Confidence:** 4

**Summary:**

The work presents a novel optimization method called HiZOO, designed for fine-tuning large language models (LLMs). HiZOO addresses this by incorporating a diagonal Hessian estimation through an additional forward pass, allowing it to act as a pre-conditioner that adjusts updates based on parameter curvatures. This approach reduces training steps and improves accuracy while maintaining efficient memory use, scaling well even for models with billions of parameters. The authors also propose HiZOO-L, a low-rank version that significantly cuts down on memory cost while preserving performance.

**Strengths:**

1. The introduction of HiZOO, a Hessian-informed zeroth-order optimizer, is original.
2. The proposed low-rank variant, HiZOO-L, which reduces memory overhead, demonstrates a approach to solving memory constraints while maintaining optimization quality.
3. The paper is well-organized, with clear sections outlining the motivation.

**Weaknesses:**

1. The code implementation assumes that  u_i Hadamard product u_i is treated as a diagonal matrix. However, this is incorrect as \( u_i u_i^T \) is an outer product in the algorithm description in the paper resulting in a rank-1 matrix. This assumption could introduce inconsistencies or inaccuracies in the Hessian estimation and parameter updates.

2. The value for `Hessian_smooth` in the implementation is set to \( 1e^{-6} \), which seems quite small. This could imply that the contribution of the Hessian information is not significant enough to make a substantial impact on convergence or stability.

3. Tables 1 and 3 present results for a range of NLP tasks, but no generation tasks are included. This limits the understanding of HiZOO’s performance in broader language modeling applications where generative capabilities are critical.

4. While Table 5 reports promising results on SST2, Figures 12 to 14 reveal that HiZOO demonstrates poor training performance and even training failure in some cases. This discrepancy raises concerns about the stability and reliability of the optimizer under different conditions.

5. Line 9 in Algorithm 2 is unclear. Calculating \( R^{-1} \) and \( C^{-1} \) involves significant computational overhead, making this step inefficient.

**Questions:**

Please see weakness.

---

> ### Author Response · Authors · 2024-11-16
> **Thank you for your valuable comments**
>
> Thank you for your thorough review and valuable comments on our manuscript. We appreciate the opportunity to address your concerns and clarify any misunderstandings.\
> **Question 1: Treatment of $uu^{\top}$ as a Diagonal Matrix**\
> [Revised answer]Firstly we would like to address the broader computational considerations. Due to the significant memory and computational overhead required to estimate the full Hessian matrix $d \times d$, we chose to compute the diagonal of the Hessian instead. This approach strikes a balance between computational feasibility and the accuracy needed for our analysis. As a result, in our code, we do indeed compute $diagonal(u_i u_i^{\top}) = [u_1^2, u_2^2, ... u_n^2]$, whose rank is not 1 and you are correct. The statement in the algorithm is included to illustrate the theoretical process of our approach. However, in the actual computation, particularly in Line 9, this step is a middle state and does not appear. It is meant purely for clarity in explaining the algorithm's underlying theory.
>
> **Question 2: Small Value of the Hessian Smooth Parameter** \
> Due to the large variance of the Hessian, with some values being extremely large, so Hessian smooth term is very small. But it is still sufficient to have a meaningful impact on the convergence and stability of the optimization process. Our experimental results demonstrate that, under the same parameter configurations, HiZOO significantly improves the convergence rate compared to other methods.
>
> **Question 3: Experiments on Generation Tasks**\
> Thank you for your insightful suggestion. We agree that including generation tasks would provide a more comprehensive evaluation of HiZOO's performance. We will conduct additional experiments on generation tasks and show the results to you as soon as possible to showcase HiZOO's capabilities in generative settings.
>
> **Question 4: some specific figures**\
> We acknowledge that sometimes HiZOO exhibits poor training performance in certain cases, it's also the same for MeZO or other zeroth-order methods. This issue arises due to the specific characteristics of some tasks, instead of the algorithm. However, HiZOO generally demonstrates better convergence behavior than MeZO under the same conditions.
>
>
> **Question 5: Computational cost of calculating R and C matrix in Algorithm 2**
>
> It 's true that this calculation involves some additional computations. However, the time required for generating these matrices is minimal compared to the overall time of one forward pass. **As shown in Table below (also Table 11 in Paper), we indicate that matrix decomposition increases only about 6\% of the total training time. Therefore, this calculation does not significantly impact the efficiency of the algorithm**.
>
> ---
> **Table: Wallclock time per step between MeZO, HiZOO, and HiZOO-L.**
> The increase in wallclock time per step for HiZOO compared to MeZO is less than 1.5 times across different model sizes. Also, HiZOO-L increases only about 6\% of the total training time than HiZOO. To avoid introducing additional overheads such as inter-GPU communication, results are measured on the same dataset (SST-2) and GPUs (80GB A100), with each result averaged over 100 steps. "BS" refers to batch size. For the relatively smaller RoBERTa-large model, we used a BS=64, while for models larger than 1B parameters, we used a BS=16.
>
> | Model         | **RoBERTa-large (350M)** | **Phi-2 (2.7B)** | **Llama3 (8B)** | **OPT (13B)**  |
> |---------------|--------------------------|------------------|-----------------|----------------|
> | **MeZO**      | 0.2092s (BS=64)          | 0.3011s (BS=16)  | 0.7471s (BS=16) | 1.1108s (BS=16)|
> | **HiZOO**     | 0.3023s (BS=64)          | 0.4486s (BS=16)  | 1.1090s (BS=16) | 1.5225s (BS=16)|
> | **HiZOO-L**   | 0.3193s (BS=64)          | 0.4851s (BS=16)  | 1.1996s (BS=16) | 1.6422s (BS=16)|
>
> ---
>
> We appreciate your constructive feedback and believe that addressing these points will strengthen our manuscript. We are committed to improving our work and will incorporate the necessary revisions to enhance its clarity and comprehensiveness.

---

> > ### Comment · Reviewer_Wjn7 · 2024-11-27
> >
> > Thanks for the authors' feedback. The authors have provided comprehensive responses addressing my comments, clarifying technical details, and proposing meaningful updates to the manuscript.
> >
> > The authors effectively addressed concerns regarding the small value of the Hessian smooth parameter. By quantifying the overhead (6% increase in training time for HiZOO-L) and contextualizing it with respect to total training time, it justifies that this additional cost is minimal.
> >
> > **Minor concern**: In the comments, the authors mentioned that \( uu^{\top} \) is not diagonal and is not a rank-1 matrix in \( \mathbb{R}^{m \times m} \). I am a bit confused. Please clarify.
> >
> > These updates enhance the manuscript’s clarity, applicability, and technical depth, so I would like to give an increased score of 6.

---

> > > ### Author Response · Authors · 2024-11-27
> > > **Thank you again for your efforts!**
> > >
> > > Dear Reviewer Wjn7, we want to extend our heartfelt thanks for taking the time to review our rebuttal and for taking the time to thoroughly review our code.
> > >
> > > About your concern, we sincerely apologize for initially misunderstanding your question and truly appreciate your effort in carefully examining our implementation. Firstly we would like to address the broader computational considerations. Due to the significant memory and computational overhead required to estimate the full Hessian matrix $d \times d$, we chose to compute the diagonal of the Hessian instead. This approach strikes a balance between computational feasibility and the accuracy needed for our analysis. As a result, in our code, we do indeed compute $diagonal(u_i u_i^{\top}) = [u_1^2, u_2^2, ... u_n^2]$, whose rank is not 1 and you are correct. The statement in the algorithm is included to illustrate the theoretical process of our approach. However, in the actual computation, particularly in Line 9, this step does not appear. It is meant purely for clarity in explaining the algorithm's underlying theory.
> > >
> > > Thank you once again for pointing this out and for your constructive comments. Your input has been invaluable in improving the clarity and correctness of our work. Feel free to ask if you have any other concerns.

---

> ### Author Response · Authors · 2024-12-03
> **Thank you for your support and we provide further insights!**
>
> Under the assumptions of gradient descent and convex optimization, alongside Assumption $
> \frac{tr(\Sigma^{1/2}H\Sigma^{1/2}) }{\|\| \Sigma^{1/2}H\Sigma^{1/2}\|\|_2} < r$, we believe there is strong potential to demonstrate that the diagonal-Hessian based preconditioning in HiZOO offers significant advantages over MeZO. Such findings would underscore the promising practical applications of our method and its effectiveness. We are willing to share further insights and are committed to enhancing this aspect of the theory, trying to ensure a more comprehensive theoretical explanation is included in the final version of the paper.
>
> Also, it is important to acknowledge that extending these advantages to scenarios involving stochastic gradient descent (SGD) or non-convex optimization presents significant challenges. To the best of our knowledge, there is few or even no existing work in the field that has successfully established improvements under these conditions. This highlights the value of our contributions within the specific scope of convex optimization and gradient descent.
>
> We sincerely hope that you will reconsider our work and, if possible, provide further support for our research.

---

### Official Review · Reviewer_MAUK · 2024-11-04

**Soundness:** 2
**Presentation:** 3
**Contribution:** 2
**Rating:** 5
**Confidence:** 4

**Summary:**

This paper proposes the Hessian-informed Zeroth-Order Optimizer (HiZOO), which achieves faster convergence than traditional zeroth-order SGD (specifically, MeZO) in LLM fine-tuning scenarios by leveraging Hessian information through a zeroth-order approach. Although HiZOO requires one additional forward pass compared to MeZO (totaling three forward passes), it demonstrates faster convergence in terms of the number of function calls. Furthermore, HiZOO outperforms MeZO in terms of model generalization for various tasks in LLM fine-tuning and the authors also provide convergence guarantee of their HiZOO method in theory.

**Strengths:**

1. The first approach that leverage the second-order information via zeroth-order construction for fine-tuning LLMs
2. Illustration of HiZOO on some test functions with superior generalization ability of the method
3. Convergence guarantee of HiZOO with their diagonal inverse Hessian estimator

**Weaknesses:**

1. The Hessian estimator from Equation (3) appears to estimate the Hessian matrix only if the matrix $\Sigma$ is close to the inverse Hessian.
However, according to Algorithm 1 in the paper, I saw that the matrix (actually, the vector) $\Sigma_0$ is initialized simply as the identity matrix, which suggests that this estimator may not accurately estimate the (diagonal) inverse Hessian. Furthermore, since the initial value is merely an identity matrix, it is difficult to consider the subsequent estimated $\Sigma_t$ values as accurate estimations of the diagonal inverse Hessian.
Instead, it seems more likely that HiZOO are estimating something positive definite.

---

2. The theoretical result seems a bit unusual.
Typically, convergence results are derived as bounds on the norm of the true gradient (i.e. the gradient evaluated on the entire training dataset) rather than the norm of the stochastic gradient.
Consequently, Theorem 3.2, which presents the current convergence result, is very confusing and may need to be revised for clarity.

---

3. In addition to the second concern, the convergence rate in Theorem 3.2 is on par with first-order methods. However, given that second-order information, such as the diagonal Hessian, is utilized—even if constructed solely from function values—there should ideally be an advantage in convergence compared to existing methods like MeZO or other vanilla SGD-based approaches. This expected benefit, however, does not seem apparent. Moreover, $\max_t \mathrm{Tr}(\Sigma_t)$ would also depend on $d$.
For example, given that Algorithm 1 initializes $\Sigma_0$ as the identity matrix, the quantity $\mathrm{Tr}(\Sigma_0)$ becomes $d$, so $\max_t \mathrm{Tr}(\Sigma_t)$ is at least on the order of the parameter dimension $d$.
If the order of $\max_t \mathrm{Tr}(\Sigma_t)$ exceeds $d$ (e.g., something like $d\sqrt{d}$), the convergence rate would actually be slower than zeroth-order vanilla SGD (ZO-SGD), which undermines the theoretical contributions of the proposed algorithm.
(To the best of my knowledge, the convergence of ZO-SGD can be derived faster than $O(d)$). Also, importantly, the advantage of leveraging second-order information should be apparent in terms of theory.

---

4. Looking at the comparison on the test function, it appears to be a comparison of just optimization from scratch rather than fine-tuning.
I'm curious about the hyperparameter settings, as it seems to perform better than Adam.
If this advantage holds, there should also be experiments demonstrating the use of zeroth-order methods not only in fine-tuning but also in pre-training.

**Questions:**

Please refer to the weaknesses.

---

> ### Author Response · Authors · 2024-11-16
> **Thank you very much for your concern about our theoretical results.**
>
> We deeply appreciate the time and effort you have invested in reviewing our manuscript. Your thoughtful comments and constructive suggestions have been invaluable in improving the quality of our work. Thank you for your insightful and detailed feedback.\
> \
> **Question 1:  $\Sigma$ is not required to be close to the inverse Hessian**\
> We apologize for any confusion caused by our previous presentation. To clarify, the Hessian matrix in our method does not require the condition that the matrix be close to the inverse Hessian, which means that:
>
> Drawing from the lemma presented in MiNES(Mirror Natural Evolution Strategies):
>
> $\frac{1}{2} \cdot \mathbb{E}_u (u^{\top} \Sigma^{1/2} H \Sigma^{1/2} u \cdot (\Sigma^{-1/2}u u^{\top} \Sigma^{-1/2} - \Sigma^{-1}))=H$
>
> where $H$ is the Hessian $\nabla^2 \mathcal{L}(\theta; \mathcal{B})$ and $\Sigma$ is a positive definite matrix.
>
> **Question 2: theoretical result of convergence**\
> Thank you for your valuable feedback. We sincerely apologize for the typo in our manuscript. We will revise Theorem 3.2 to reflect this correction for clarity, shown as below:
>
> Let the descent direction $g_\mu(\theta_t)$ defined as:
>
> $g_\mu(\theta_t)
>         =
>         \sum_{i=1}^b \frac{\mathcal{L}(\theta_t + \mu \Sigma_t^{1/2} u_i; \mathcal{B}_t) - \mathcal{L}(\theta_t - \mu \Sigma_t^{1/2} u_i;\mathcal{B}_t)}{2b\mu} \Sigma_t^{1/2}u_i.$
>
> Based on Assumption, if the update rule for $\theta$ is $\theta_{t+1} = \theta_t - \eta g_\mu(\theta_t)$ for a single step, then it's established that:
> $ \mathbb{E} \left[\mathcal{L}(\theta_{t+1}) \mid \right]
>     \leq \mathcal{L}(\theta_t) - \frac{\eta_t}{4} \|\nabla \mathcal{L}(\theta_t)\|_{\Sigma_t}^2
>     + 2\eta_t^2L\left( \mathrm{tr}(\Sigma_t) +\beta_u \right)\sigma^2 + O(\mu^2).$
>
> Furthermore, given iteration number $T$, we choose the step size $\eta = \frac{1}{8\sqrt{T}L(\max_t\mathrm{tr}(\Sigma_t) +\beta_u)}$ and take $\theta_{\mbox{out}} = \theta_j$ with $j$ uniformly sampled from $\{1, \dots, T\}$.
> Then, we have
>
> $\mathbb{E} \left[\|\nabla \mathcal{L}(\theta_{\mbox{out}})\|^2\right] \leq \frac{32L\left( \max_t \{\mathrm{tr}(\Sigma_t)\} +\beta_u \right)
>  \mathcal{L}(\theta_1) - \mathcal{L}(\theta_*))}{\sqrt{T}\beta_\ell }  + \frac{\sigma^2}{T^{3/2} \beta_\ell} + O\left(\mu^2\right),$
> where $\mathcal{L}(\theta_* )$ minimizes the function $\mathcal{L}(\theta;)$. The above equation shows that as $T\to \infty$, HiZOO can converge to the stationary point.\
>
> **Question3** Theoretical concern\
> In fact, in the realm of non-convex optimization, it is difficult to show that a Newton-like algorithm with  approximate Hessian matrices can help to achieve faster convergence rate. And we admit that our convergence analysis does not provide theoretical support that our method can achieve a fast convergence rate. Our theory mainly is used to show that our algorithm can converge to some stationary point.\
> We have provided an updated convergence proof for your reference. Due to its length, we have directly updated the PDF file, specifically in Line 763-884.
>
> **Question4: apply HiZOO in pre-training**\
> Thank you for your insightful comment. Indeed, there have been studies that have applied zeroth-order optimizers for training neural networks from scratch, eg.[DeepZero: Scaling up Zeroth-Order Optimization for Deep Model Training]. However, due to the high computational cost associated with training from scratch, in this work we focus on fine-tuning. In the future, we are willing to explore using HiZOO for pre-training.\
> Additionally, in our test functions, we set the exactly same hyper-parameters for Adam, MeZO and our method. For example, in function: $f(x,y) = 10(x-1)^2*(2x^2+x+1)+2(y-4)^2$, we set the same start point$(-1,2)$, end point(1,4), learning rate $10^{-3}$, scaling factor $10^{-6}$, and we can see that in the same setting, our method performs better than MeZO.

---

> ### Author Response · Authors · 2024-11-27
> **Hope for further discussion or feedback. Thanks for your time!**
>
> Dear Reviewer MAUK, we greatly appreciate the time and effort you have dedicated to reviewing our submission.
>
> As part of the ongoing discussion period, we have provided detailed responses to the points raised in your review. We would sincerely appreciate any additional feedback or clarifications you may have. With only a few days remaining in the discussion period, we are eager to ensure a productive and collaborative exchange.
>
> Thank you once again for your valuable input and consideration.

---

> ### Comment · Reviewer_MAUK · 2024-11-28
> **Response to Authors**
>
> Thank you to the authors for their responses.
>
> However, from a theoretical perspective, the convergence of HiZOO cannot outperform vanilla SGD due to the trace term, which fails to justify the necessity of using this inverse Hessian estimator. Additionally, after reading the MiNES paper, the proposed estimator appears to estimate not the inverse Hessian but rather something closer to the inverse Fisher matrix. While Fisher and Hessian may share similarities in estimating curvature information, they cannot be considered equivalent in large language models (LLMs) that do not use ReLU activations. This raises a fundamental question about whether HiZOO can truly be considered a Hessian-informed method. Moreover, even if it is correct to say that it estimates the inverse Fisher matrix, works like [1] have shown that natural gradient descent (using K-FAC, approximate NGD) achieves linear convergence. Thus, HiZOO should at least provide theoretical insights or intuition in this regard.
>
> For these reasons, I will maintain my current score
>
> [1] Fast Convergence of Natural Gradient Descent for Overparameterized Neural Networks, G. Zhang et al., NeurIPS 2019.

---

> ### Author Response · Authors · 2024-11-29
> **Thank you for your questions and we will clarify your concerns!**
>
> First and foremost, we sincerely appreciate your effort in raising new questions for further exploration. We will address each of your concerns in detail below, and we hope our responses will earn your understanding and approval.
>
>  (Also we want to kindly remind you that this work primarily focuses on experimental contributions. As a result, we conduct lots of experiment result to show the performance of our method and do not extensively delve into theoretical analyses.)
>
> **Question:convergence of HiZOO cannot outperform vanilla SGD due to the trace term, fails to justify the necessity of inverse Hessian estimator.**
>
> **Answer:**
> Thank you for your insightful comment on the convergence properties of HiZOO. As indicated in the theoretical analysis of MeZO[1] (HiZOO is an improvement based on MeZO), the loss decrease for MeZO can be expressed as:
>
> $\mathbb{E}[L(\theta_{t+1}) | \theta_t] - L(\theta_t) \leq \frac{1}{\gamma} \left[-\eta_{SGD} \|\nabla L(\theta_t)\|^2 + \frac{1}{2} \eta_{SGD}^2 \ell \cdot \mathbb{E}[\|\nabla L(\theta; \mathcal{B})\|^2] \right],$
>
> where $\gamma^{-1} = O(n/r)$, $n$ denotes the number of queries per iteration, and $r$ is the effective rank of the Hessian. For MeZO/HiZOO, $n = 1$, which implies that its convergence rate depends solely on $r$, the effective rank of the Hessian, rather than $d$, the parameter dimension, as in traditional SGD.
>
> In traditional zeroth-order SGD, the efficiency is scaled by $d$, as the loss decrease is proportional to $1/d$ due to the dimensionality of the parameter space. However, since $r$ is much smaller than $d$, the factor $1/r$ in MeZO and HiZOO results in a faster loss decrease compared to $1/d$ in SGD. A detailed analysis of this can be found in the convergence analysis section of MeZO.
>
> This distinction is significant because, as shown in Adahessian[3], Sophia[4], Shampoo[5], employing curvature information (Hessian approximations) enables algorithms to converge faster than first-order methods like SGD, especially in high-dimensional optimization problems. Specifically, the effective rank $r$ is typically much smaller than the parameter dimension $d$, owing to the low-rank property of the Hessian matrix observed in many practical machine learning scenarios.
>
> Moreover, HiZOO improves upon MeZO by introducing second-order information through preconditioning methods that estimate the diagonal of the Hessian.
>
> **Question:HiZOO can truly be considered a Hessian-informed method or Fisher-informed method?**
>
> **Answer: HiZOO is definitely a Hessian-informed method.**
> Thank you for raising this insightful question. Regarding the convergence properties of HiZOO, we would like to emphasize that our method indeed estimates the inverse Hessian matrix, as demonstrated in the theoretical analysis of MiNES[2]. Specifically, Lemma 11 in MiNES establishes that the sequence $\Sigma_k^{-1}$, generated by their algorithm, satisfies:
>
> $\|\Sigma_k^{-1} - \Pi_{\mathcal{S}} (\nabla^2 f(\mu^*))\|^2 \leq \frac{C}{k},$
>
> where $\nabla^2 f(\mu^*)$ is the Hessian matrix of the loss function $f$ at the optimal point $\mu^*$. The projection $\Pi_{\mathcal{S}}$ removes the extreme eigenvalues (both maximum and minimum) of the original Hessian matrix $\nabla^2 f(\mu^*)$, effectively regularizing it within a spectral range defined by $\mathcal{S}$. When $\mathcal{S}$ is chosen to be sufficiently large, $\Pi_{\mathcal{S}} (\nabla^2 f(\mu^*))$ becomes equivalent to $\nabla^2 f(\mu^*)$ itself, as the projection no longer affects the spectral properties of the Hessian.  This result clearly demonstrates that $\Sigma_k^{-1}$ approximates $\nabla^2 f(\mu^*)$ as $k \to \infty$, at a convergence rate of $O\left(\frac{\log k}{k}\right)$.
>
> While the MiNES paper also discusses the Fisher matrix $F_\mu$ in the context of natural gradient descent, it is important to note that this section is unrelated to our analysis. HiZOO focuses solely on the inverse covariance matrix $\Sigma_k^{-1}$, which approximates the Hessian matrix and does not involve the Fisher matrix. The distinction is particularly evident in the update rules for $\Sigma_k$, which are derived based on Hessian-specific properties rather than Fisher information.
>
>
> [1]Malladi, Sadhika, et al. "Fine-tuning language models with just forward passes." Advances in Neural Information Processing Systems 36 (2023): 53038-53075.
>
> [2]Ye, Haishan and Tong Zhang. “Mirror Natural Evolution Strategies.” ArXiv abs/1910.11490 (2019): n. pag.
>
> [3]Yao, Z., Gholami, A., Shen, S., Keutzer, K., & Mahoney, M.W. (2020). ADAHESSIAN: An Adaptive Second Order Optimizer for Machine Learning. ArXiv, abs/2006.00719.
>
> [4]Liu, H., Li, Z., Hall, D.L., Liang, P., & Ma, T. (2023). Sophia: A Scalable Stochastic Second-order Optimizer for Language Model Pre-training. ArXiv, abs/2305.14342.
>
> [5]Anil, R., Gupta, V., Koren, T., Regan, K., and Singer, Y. Scalable second order optimization for deep
> learning. arXiv preprint arXiv:2002.09018, 2020.

---

> > ### Author Response · Authors · 2024-11-29
> > **Thank you for your questions and we will clarify your concerns!**
> >
> > **Question:natural gradient descent achieves linear convergence. Thus, HiZOO should provide theoretical insights or intuition in this regard.**
> >
> > **Answer:**
> > Thank you for raising this point. As discussed earlier, the convergence rate of both MeZO and HiZOO is determined by the effective rank $r$ of the Hessian, which is significantly smaller than the parameter dimension $d$ in most practical scenarios. This inherent advantage enables HiZOO to converge faster than methods like SGD, whose efficiency scales with $d$.
> >
> > Additionally, HiZOO leverages diagonal precondition to further accelerate the estimation process. Diagonal preconditioning has been extensively used in many works to enhance optimization efficiency, including well-known optimizers such as Adahessian[1], Sophia[2], Shampoo[3]. These methods employ similar diagonal adjustments to improve convergence speed and stability in high-dimensional settings.
> >
> > [1]Yao, Z., Gholami, A., Shen, S., Keutzer, K., & Mahoney, M.W. (2020). ADAHESSIAN: An Adaptive Second Order Optimizer for Machine Learning. ArXiv, abs/2006.00719.
> >
> > [2]Liu, H., Li, Z., Hall, D.L., Liang, P., & Ma, T. (2023). Sophia: A Scalable Stochastic Second-order Optimizer for Language Model Pre-training. ArXiv, abs/2305.14342.
> >
> > [3]Anil, R., Gupta, V., Koren, T., Regan, K., and Singer, Y. Scalable second order optimization for deep learning. arXiv preprint arXiv:2002.09018, 2020.

---

> ### Comment · Reviewer_MAUK · 2024-12-03
> **Response to Authors**
>
> I appreciate the authors' response.
>
> However, the local effective rank condition (presented in MeZO) could also be applied to the analysis for HiZOO. Also, the proof technique in current version of HiZOO does not give theoretical superiority over the MeZO even with the local effective rank condition. So, I think that the authors should improve this part for future direction.
>
> I also understand this work does more focus on the empirical studies, so I raised the score up to 5.
> But I strongly recommend that the authors should revise the theory related to trace of estimators.

---

> > ### Author Response · Authors · 2024-12-03
> > **Thanks for your constructive suggestions and your support in raising the score**
> >
> > Thank you again for your valuable feedback and for recognizing the empirical contributions of our work. **We appreciate your suggestion to enhance the theory related to the trace of estimators and consider the application of the local effective rank condition in HiZOO**.
> >
> > **We are actively working on this improvement and will try hard to provide a detailed proof with as much depth as possible before the rebuttal phase concludes.** Your insights are critical, and we hope you can consider our updated analysis during the reviewer discussion phase.
> >
> > Thanks again for your constructive suggestions and your support in raising the score

---

### Official Review · Reviewer_RuKr · 2024-11-04

**Soundness:** 3
**Presentation:** 2
**Contribution:** 2
**Rating:** 6
**Confidence:** 3

**Summary:**

Motivated by the limited optimization effectiveness of zeroth-order optimizers in deep learning, this manuscript leverages a diagonal Hessian to enhance the optimization quality. Though introducing second-order curvature information to aid the optimization is standard, the manuscript is quite comprehensive: it builds methods step-by-step, provides a theoretical justification, and verifies the effectiveness across various model architectures and datasets.

**Strengths:**

* The manuscript studies an interesting problem: efficient and effective zeroth-order fine-tuning. The manuscript has a clear motivation in the introduction part, with a thorough step-by-step explanation in Section 3.3. Extensive empirical studies consider evaluating both encoder-decoder and decoder-only neural architectures on the GLUE benchmark over several baseline methods. The evaluation uses the number of forward passes (which is very good) and discusses memory usage and time efficiency.
* The manuscript has some preliminary convergence analysis.

**Weaknesses:**

1. The convergence analysis part can be strengthened. It would be great if the analysis could cover the case of MeZO (and other zeroth-order algorithms) and carefully explain the gain introduced by the diagonal Hessian. See some examples in [1], where the query complexity can be discussed and compared.
2. The manuscript structure can be improved. E.g., some detailed derivates in Sec 3 can be simplified, and the definition of three test functions in Sec 3.5 can be moved to the main text.
3. Extending to other LLM SFT datasets: it would be great if the manuscript could verify the effectiveness of HiZOO on some SFT datasets.

### Reference
[1] Stochastic Two Points Method for Deep Model Zeroth-order Optimization, https://arxiv.org/abs/2402.01621.

**Questions:**

NA

---

> ### Author Response · Authors · 2024-11-16
> **The reference paper you provide is of significant value and inspiration for our work.**
>
> We sincerely appreciate the time and effort you have dedicated to reviewing our manuscript. Your valuable comments and suggestions are instrumental in enhancing the quality of our work. Thank you for your insightful feedback.\
> \
> **Question 1**
> Thank you for your suggestion and this reference paper provides a very solid theoretical introduction, which is worth learning for us.
> By Theorem 3.2, we can obtain a query complexity $\left(\frac{\mathrm{tr}(\Sigma)}{\beta_\ell}\right)^2\cdot\frac{1}{\epsilon^4}$ for our method.
> Our query complexity depends on $\epsilon^{-4}$ while the reference paper depends $\epsilon^{-2}$, our baseline MeZO is also  $\epsilon^{-4}$. This is because our method samples function (namely Stochastic Gradient Descent) while the this paper uses all functions per iteration (namely Gradient Descent). So we could not conduct direct comparison. However, we will cite this reference paper and built upon their methodology in our work.
>
> **Question 2**
> We will simplify the detailed derivations in Section 3 as recommended. Additionally, we will move the definitions of the three test functions in Section 3.5 to the main text to enhance clarity and readability.
>
> **Question 3**
> Thank you for pointing out the importance of extending our experiments. On Roberta-large we conduct the few-shot learning experiment and on OPT, Llama, Phi models we have conducted the full-shot learning, which have included the supervised situations. Also, we will consider including more datasets or models to evaluate our method.

---

> ### Author Response · Authors · 2024-11-27
> **Thank you a lot for your support!**
>
> Dear Reviewer RuKr, we greatly appreciate the time and effort you have dedicated to reviewing our submission.
>
> We have read this paper [Stochastic Two Points Method for Deep Model Zeroth-order Optimization], https://arxiv.org/abs/2402.01621. This paper is truly impressive, with solid theoretical foundations and remarkable contributions. I greatly admire this work and see it as a valuable resource for my learning.
>
> As part of the ongoing discussion period, we have provided detailed responses to the points raised in your review. We would sincerely appreciate any additional feedback or clarifications you may have. With only a few days remaining in the discussion period, we are eager to ensure a productive and collaborative exchange.
>
> Thank you once again for your valuable input and consideration.

---

> > ### Comment · Reviewer_RuKr · 2024-11-28
> > **Thank you for the feedback**
> >
> > Thank you for the feedback. I will maintain my current rating.

---

### Meta-Review · Area_Chair_YJw4 · 2024-12-20

**Metareview:**

This paper proposes to precondition MeZO with an approximated, EWM-denoised diagonal Hessian, which is computed via an additional forward pass per step. The authors demonstrate superior convergence performance, and prove convergence of their method, HiZOO in the L-smooth but nonconvex setting. The strengths of the paper are the strong empirical performance and thorough evaluation of their fine tuned models. I also appreciate that the authors directly addressed the obvious additional memory requirements of storing the length-d Hessian diagonal. The proof of convergence in the L-smooth but non-convex setting.

**Additional Comments On Reviewer Discussion:**

The largest remaining weakness from the rejection voting reviewer seems to be the theoretical result in the paper not being substantially better than what is achievable with first order methods (some of the other concerns, e.g. that referenced work analyzes natural gradient descent and not Newton's method does not seem to hold). From my perspective, because the authors are only assuming L-smoothness of the objective function, I don't think it's reasonable to expect better convergence rates here -- in general, proving faster convergence of second order methods on nonconvex functions I think requires stronger assumptions than this (e.g., smoothness of the Hessian).

Frankly, I don't think that an LLM fine-tuning paper is the appropriate place to litigate theoretical gaps in first versus second order method convergence rates, which is a topic worthy of study in its own right. From an empirical perspective it's not unreasonable to conjecture that including some curvature measure in the optimization procedure could improve performance, and indeed it appears to do so here.

---

### Decision · Program_Chairs · 2025-01-22

Accept (Poster)